# Differentiable Weightless Neural Networks: A Benchmark on Performance and Robustness

## Abstract

Miniaturizing Machine Learning (ML) models to operate accurately in resource-constrained environments may improve the intelligence of everyday objects. Applications abound across the Internet of Things (IoT), for personal medical devices, and for consumer electronics such as smartphones and augmented reality glasses. Differentiable Weightless Neural Networks (DWNN), such as differentiable Logic Gate Networks (LGN) and Look Up Table networks (LTN), represent a class of models which accelerate ML inference by orders of magnitude while retaining high predictive accuracy. While small-scale LGNs and LTNs already expedite model inference and reduce resource usage, performance and robustness benchmarks are not well reported which hinders the development of large architectures suitable for real-world applications. This paper fills this gap by comparing LGNs, LTNs, Multi-Layer Perceptrons (MLPs), and their convolutional counterparts on the basis of test accuracy, training time, and robustness to noise across key model and training variations. We introduce the Look-Up Table Convolutional Network (LTCNN), which reduces training time by 2–3X compared to prior logic-based convolutions. By benchmarking over 4,000 models, we identify critical scaling limitations: unlike standard CNNs, increasing DWNN parameter counts tends to exacerbate brittleness to environmental noise rather than improving generalization. Furthermore, while employing learnable interconnects improves accuracy by 6–20%, it incurs a significant computational penalty. These results quantify the distinct trade-offs of weightless architectures, highlighting the need for training strategies to scale DWNNs for real-world applications.

## 1 Introduction

As the introduction of backpropagation (Rumelhart et al., 1986) inspired a race to develop neural network architectures like deep neural networks (Fukushima, 1980), convolutional neural networks (CNN) (LeCun et al., 1989), recurrent neural networks (Elman, 1990), and modern-day transformers (Vaswani et al., 2017), progress in Differentiable Weightless Neural Networks (DWNN) reveal a similar story in scaling the capabilities of available model types. For example, differentiable LGNs, a type of WNN using networks of combinatorial logic gates, improve model inference speeds by several orders of magnitude on Field Programmable Gate Arrays (FPGA) while maintaining high predictive accuracy on simple benchmarks (Petersen et al., 2022). In the following years, convolutional (Petersen et al., 2024) and recurrent (Bührer et al., 2025) LGNs were introduced, and differentiable LTNs later improved inference speeds while reducing the circuit area of learned logic on FPGAs (Susskind, 2024; Andronic & Constantinides, 2025). These DWNNs continue to improve power efficiency, latency, and lightweight resource usage across robotics, transportation, healthcare, and agriculture (Abduljabbar et al., 2019; Eli-Chukwu, 2019; Jiang et al., 2017; Rybczak et al., 2024), where the European Council for Nuclear Research (CERN) uses WNNs to manage petabytes of information per second in data collection (Jia et al., 2024; Boggia et al., 2025).

While researchers have demonstrated that DWNNs can both achieve rapid inference and high predictive capabilities in image datasets like CIFAR10 (Petersen et al., 2022; Bacellar et al., 2024; Krizhevsky et al., 2009b), DWNNs are not yet scalable–they struggle on tasks like CIFAR100 (Rüttgers et al., 2025; Krizhevsky et al., 2009a) and ImageNet (Brändle et al., 2025; Deng et al., 2009) which are precursory baselines for image classification in real-world environments. Some obstacles, like longer training times (Petersen et al., 2022), poor network gradients (Petersen et al.,

Figure 1: Overview of the paper, outcomes, and expected impact

2024), and learning model interconnects (Bacellar et al., 2024) hinder DWNN scaling, though detailed benchmarks are unavailable. Other model properties, like robustness to noise and the impact of encoding schemes on performance, are not reported for DWNNs and are assessed from prior studies on alternate model architectures (Kappaun et al., 2016). Benchmarking DWNN properties against traditional Deep Neural Networks (DNN) quantifies explicit performance gaps, thereby highlighting key directions for DWNN research. This work pursues such benchmarks to show directions for continued effort which may help in scaling DWNNs to real-world applications, and releases the first open-source software with convolutional DWNNs to help democratize scaling pursuits.

In this benchmark, we prioritize studies on DWNN training rather than model deployment. Prior works evaluate Binary Neural Networks (Qin et al., 2020), Quantized Neural Networks (Guo, 2018), and the performance of models on FPGAs (Petersen et al., 2022; Bacellar et al., 2024), so we complement existing work with novel training comparisons. Justifications include: (1) training remains a primary bottleneck limiting DWNN scalability, (2) this study remains hardware-agnostic and broadly applicable across devices, and (3) it complements ongoing work on LGNs and LTNs which have only recently been evaluated beyond small-scale tasks (Yousefi et al., 2025; Brändle et al., 2025). This perspective emphasizes that deployment constraints on edge devices (e.g. timing, resources, model size, power) are secondary if DWNNs cannot first be trained to perform real-world tasks. Toward this end, our study characterizes training and performance metrics for feed forward and convolutional variants of LGNs, LTNs, and DNNs. The goal is to inspire research toward novel DWNN architectures and training methodologies which may scale to real-world applications. Figure 1 overviews the work in the following sections. These sections introduce the relevant background and study novelty (Section 2), the research methodology (Section 3), and benchmark results (Section 4) before providing concluding remarks alongside directions for future research (Section 5).

## 2 BACKGROUND AND NOVELTY

While DWNNs surpass traditional WNNs and prior Look-Up-Table (LUT)-based models on several real-world benchmarks, including edge-deployed workloads and CIFAR10 image classification, challenges in training and scaling these models remain (Bacellar et al., 2024). Traditional WNNs, like Random Access Memory (RAM) nodes, Probabilistic Logic Neurons (PLN), and Sparse Distributed Memory (SDM) use binary interconnects (i.e. wires) between explicit hardware resources on compute devices rather than weighting nodal connections with multiplicative factors which can be computationally expensive for small-scale devices. Differentiable LGNs and LTNs are two DWNNs, which remain unique as they may be trained using backpropagation rather than reinforcement learning, adaptive resonance theory, or other statistical methods (Ludermir et al., 1999; Kim, 2023). LGNs were introduced by Petersen et al. (2022), and LTNs (originally called Differentiable Weightless Neural Networks, or DWNs) were published later by Bacellar et al. (2024). Given the publication timeline, LGNs have been demonstrated in applications ranging from music generation to circuit synthesis, cellular automata, and efficient signal processing for ECG data (Clester, 2025; Zhou et al., 2025; Feng et al., 2024; Miotti et al., 2025) while remaining uniquely interpretable (Wormald et al., 2025; Yue & Jha, 2025). LTNs promise to improve model latency and memory requirements while improving accuracy, though fewer derivative works are available given their recency. LGNs and LTNs are prioritized in this work for their novelty, superior inference speeds on FPGAs, and reduced resource usage. The following model comparisons clarify this study's novelty:

*(1) Learned Hardware Components*: The fundamental difference between LGNs (Petersen et al., 2022) and LTNs (Bacellar et al., 2024) is the use of logic-gates and look-up-tables, respectively. Many FPGAs use a logic fabric composed of 4-input and 6-input look-up-tables (or generally, $n$-input), meaning that LTNs tailored to the appropriate look-up-table size can map directly to the hard-

ware resources on FPGAs. Because LGNs learn two-input logic gates, the synthesis of hardware-specific kernels may result in less efficient hardware usage compared to LTNs (Bacellar et al., 2024).

*(2) Learnable Parameters and Interconnects*: While LGNs learn to use one of 16 logic gates per node across a predefined network scaffold (i.e. the layer interconnects are fixed), LTNs adjust look-up-table entries while simultaneously learning the interconnects between layers. In learning interconnects, LTNs search across a larger set of architectures than the original LGN. This discrepancy may have lead to unfair comparisons between LTNs and LGNs in the original work. While numerous publications emerged in 2025 to efficiently learn inter-layer connections in LGNs (Mommen et al., 2025; Kresse et al., 2025; Yousefi et al., 2025), **a direct comparison between LTNs and LGNs under comparable mapping strategies was not found by the authors until this work.**

*(3) Toward Larger DWNN Architectures*: While LTNs improve latency and reduced memory utilization compared to LGNs, larger architectures were not built using LTN primitives until recent years (Jadhao et al., 2025). In comparison, both recurrent (RDDLGN) and convolutional[1] (LGCNN) architectures of the LGN are demonstrated (Petersen et al., 2024; Bührer et al., 2025). **Filling this gap, our work introduces the LTCNN and compares it against the LGCNN for the first time.**

*(4) Benchmarking DWNN Performance Characteristics*: Both LGNs and LTNs are tested across varied datasets and study types, though discrepancies between the two studies lead to inconsistent comparisons, specifically for: (1) the difference in training time per model architecture, (2) the robustness of each model to noise, (3) the impact of data augmentation on model performance, (4) performance changes when varying encoding schemes and bit depths. **This work compares LGNs and LTNs on these bases to identify help streamline future research efforts.**

*(5) Publicly Available Software*: While the original LGN and LTN studies released source code (Petersen et al., 2022; Bacellar et al., 2024), open-source implementations for convolutional architectures are lacking. The LGCNN implementation remains unreleased (Petersen et al., 2024), and the LTCNN is a novel contribution of this work. **We therefore introduce `wn`$^2$`Architect` to democratize research into large-scale DWNN architectures for real-world applications.**

## 3 METHODOLOGY

To realize the novelty in Section 2, this study benchmarks model performance characteristics when varying key aspects of the training pipeline: data augmentation, input encodings, and model scale and structure. Identifying performance gaps between DNNs and DWNNs at each stage of training stage helps identify weaknesses along the full pipeline. Figure 2 illustrates five studies benchmarking performance gaps when varying data augmentation ($\Delta D$), the bit depth and encoding scheme ($\Delta B$ & $\Delta E$), model type and scale ($\Delta S$), and the use of learnable mappings ($\Delta M$).

Each study was performed on MNIST (Deng, 2012), FASHIONMNIST (Xiao et al., 2017), and CIFAR10 (Krizhevsky et al., 2009b). These image-based datasets each contain ten classes of small images[2]. The analysis was limited to image datasets as DWNN performance struggles in real-world image classification tasks (Rüttgers et al., 2025; Krizhevsky et al., 2009a; Deng et al., 2009), though LGNs and LTNs have achieved state-of-the-art (SOTA) performance for MNIST and FASHIONMNIST (i.e. hand written characters and clothing items) while performing worse on CIFAR10. Model training consistently used an $80/20$ split across training and testing data, using 5-fold cross validation and a batch size of 64. Models were trained using the Adam optimizer with early stopping with a patience of 10 epochs. DNN, LGN, and LTN-based models varied the learning rate per Table 1 following the initial studies (Petersen et al., 2022; 2024; Bacellar et al., 2024). A learning scheduler decreased the learning rate with a decay rate of $\gamma = 0.95$ per epoch. The temperature $\tau$ used in the GroupSum for the LGN and LTN models was held constant at $\tau = 10$ for the feed-forward variants, and followed the relationship $\tau = \sqrt{n_f/C}$ where $n_f$ is the number of nodes on the final layer, and $C = 10$ is number of classes. This strategy follows recommendations from (Petersen et al., 2024), and resulted in values of $\tau$ between 10-35, which is comparable to the initial works. Unless otherwise stated, models are trained using random network interconnects, a constant bit depth ($b = 2$) for

---

[1]Note the LGCNN and LTCNN model types formally use a sliding window rather than convolution, though the word "convolution" is used in this text to be consistent with existing literature (Petersen et al., 2024).

[2]Images from CIFAR10 are 32x32 pixels and have three color channels, while images from the other datasets have 28x28 pixels across a single color channel.

quantized input values, and by excluding data augmentation. While prior works use thermometer encodings (Bacellar et al., 2024; Kappaun et al., 2016; Carneiro et al., 2015) we instead use standard quantization (see Appendix B) to explore whether DWNNs can leverage compact representation requiring fewer bits per value–an important advantage for reducing bandwidth when scaling to larger images. Data augmentation was not applied during training in order to evaluate the model's robustness to occlusions and salt-and-pepper (S&P) noise at test time. This approach isolates the model's inherent robustness noise and permits studies on how data augmentation changes robustness.

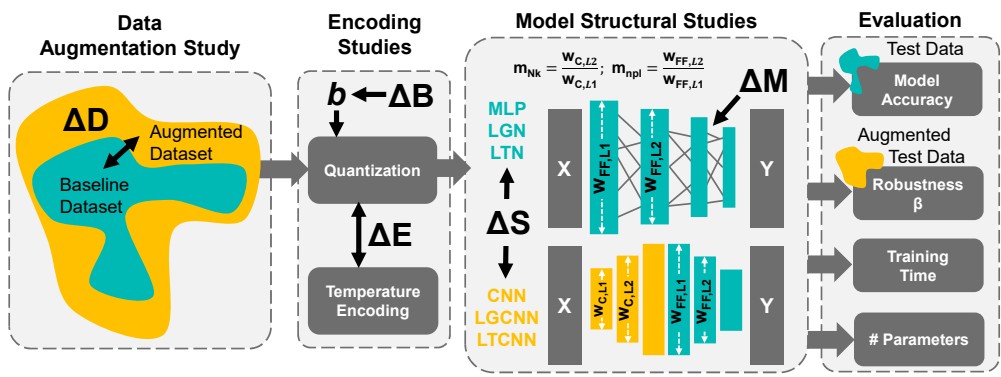

Figure 2: Overview of the studies performed (i.e. $\Delta D$, $\Delta B$, $\Delta E$, $\Delta S$, $\Delta M$), illustrating changes across data augmentation, input encodings, model structural elements, and evaluation metrics

Table 1: Parameter ranges per model type, written as [ start, end; increment ], showing the layer count ($L_f$), nodes per layer ($npl$), and $npl$ decay rate ($m_{npl}$) for feed forward models. CNN architectures use $L_c$ convolutional layers followed by a feed-forward network, use $N_k$ kernels for the first convolutional which increases by $m_{npl}$ per convolution layer. The kernel size ($k$), pool size ($N_{pool} = 2$), padding ($p = 1$), and stride ($N_s = 1$) are constant per layer. Figure 2 defines $m_{npl}$ and $m_{N_k}$. LTN-based models use $n$-input look-up-tables, bounded as [2,6;1] during architecture search

| Model | $\eta$ | $L_c$ | $N_k$ | k | $m_{N_k}$ | Q | $L_f$ | npl | $m_{npl}$ |
|---|---|---|---|---|---|---|---|---|---|
| MLP | 0.001 | – | – | – | – | – | [1, 16; 1] | [3, 2560; 1] | [0.5, 1.0; 0.1] |
| LGN | 0.01 | – | – | – | – | – | [1, 16; 1] | [10, 10000; 10] | [0.5, 1.0; 0.01] |
| LTN | 0.01 | – | – | – | – | – | [1, 16; 1] | [10, 5000; 10] | [0.5, 1.0; 0.1] |
| CNN | 0.001 | [1, 8; 1] | [1, 16; 1] | [2, 5; 1] | [1.0, 3.0; 0.25] | {–, 1, 2, 4, 8} | [1, 8; 1] | [16, 256; 1] | [0.1, 1.0; 0.01] |
| LGCNN | 0.02 | [1, 8; 1] | [1, 16; 1] | [2, 5; 1] | [1.0, 3.0; 0.25] | {–, 1, 2, 4, 8} | [1, 8; 1] | [10, 10000; 10] | [0.5, 1.0; 0.01] |
| LTCNN | 0.02 | [1, 8; 1] | [1, 16; 1] | [2, 5; 1] | [1.0, 3.0; 0.25] | {–, 1, 2, 4, 8} | [1, 8; 1] | [10, 5000; 10] | [0.1, 1.0; 0.01] |

$\Delta S$ - *Parameter Scaling Study*: This study seeks to quantify how MLP, LGN, and LTN primitives scale to more complex model types by (1) studying feed forward and convolutional architectures, and (2) increasing the parameter count ($\#P$). Five target model sizes were studied (i.e. $\#P_{target}$ = $10^{4.0}$, $10^{4.5}$, $10^{5.0}$, $10^{5.5}$, and $10^{6.0}$) mirroring the parameter range from Petersen et al. (2022). Twenty-five architectures were generated per model type using an evolutionary algorithm per Appendix A, where these 750 models define $P_{\Delta S}$ to help study how models scale across size and form.

$\Delta D$ - *Data Augmentation Study*: This study asks whether DWNNs and DNNs benefit equally from data augmentation. Here, LGNs, LTNs, and MLPs from $P_{\Delta S}$ for $\#P_{target} \in \{10^{4.5}, 10^{5.0}\}$ were retrained with data augmentation (i.e. 0-10% of pixels have salt-and-pepper noise; 4-8 pixel rectangular occlusions are inserted) to measure changes in accuracy and robustness to noise. These additional 150 models, combined with the 150 without augmentation, constitute $P_{\Delta D}$.

$\Delta B$ & $\Delta E$ - *Input Encoding Studies*: The $\Delta B$ study determines which model primitives more effectively learn from additional information given by increasing bit-depth or $b$, and complements the $\Delta E$ study on quantization and temperature encodings. $\Delta B$ and $\Delta E$ retrain five MLPs, LGNs, and LTNs from each $\#P_{target}$ using temperature and quantized encoding methods for $b \in \{1, 2, 4\}$ to compare the resulting test accuracy. These 450 models define $P_{\Delta B}$ and $P_{\Delta E}$.

$\Delta M$ - *Learnable Mapping Study*: This study seeks to benchmark the gap between model accuracy, parameter count, and training time for LGNs and LTNs when learning network interconnects. The LGN and LTN networks from $P_{\Delta S}$ where $\#P_{target} \in \{10^{4.0}, 10^{4.5}\}$ were trained with learnable mappings. These 100 models and variations with random mappings from $P_{\Delta S}$ are called $P_{\Delta M}$.

Each study was evaluated using a subset of the following metrics focused on model training and performance: (1) training time per epoch, (2) test accuracy, (3) parameter count, (4) robustness, and (5) bit-depth sensitivity. Here, robustness defines the decay rate of test accuracy, or $\beta$, in response to noise as defined in Appendix B. The bit-depth sensitivity characterizes the change in test accuracy when increasing the bit depth, or $\Delta Accuracy / \Delta b$, as defined in Appendix D.

The full study represents 1450 architectural variations trained per dataset, resulting in 4350 saved models. Experiments were conducted on NVIDIA L4 GPU nodes, each with a cyclic allocation of 128GB CPU and 24GB GPU for processing. In total, it took >67 days CPU time to run all experiments, run across a 35-node parallel system.

## 4 RESULTS AND DISCUSSION

A series of performance gaps are identified in analysis which illustrate key directions for improving DWNNs. Studies first confirm generally understood principles regarding DWNNs – there are gaps in test accuracy and training time when comparing DNN and DWNNs with the same parameter count. However, more subtle observations arise upon closer inspection, as per the following discussion:

$\Delta S$ - *Improved LTCNN Scalability*: The LTCNN introduced in this work reduces the training time by 2-3X compared to the LGCNN while generally improving model accuracy. These outcomes are expected given the higher accuracy and reduced training times for LTNs (Bacellar et al., 2024), though provides novel empirical evidence that certain DWNN primitives scale better to convolutional architectures while *revealing the benefits of continued research on efficient DWNN primitives*.

$\Delta S$ - *Gap in LGCNN and LTCNN Scalability Compared to CNNs*: One benefit of CNNs is their ability to achieve higher performance than MLPs in a comparable training time, though these benefits are not realized when scaling to LGCNNs and LTCNNs. For example, Table 2 compares the difference between the mean and maximum test accuracy per model type, shown as $\Delta\mu_{<model>}$ and $\Delta max_{<model>}$, respectively. Here, CNNs generally increase the mean test accuracy compared to MLPs[3], especially for CIFAR10. In comparison, LGCNNs and LTCNNs tend to increase the maximum test accuracy compared to LGN and LTN networks, but often decrease the mean accuracy. Considering model architectures were designed using a semi-random search (see Appendix A), these observations indicate that the probability of performant LGCNNs and LTCNNs is lower than for CNNs under the current architecture design (see Appendix E). Another key gap is in training time (see Table 3), which shows that LGCNN and LTCNN networks can increase training times by 2.8-10X, respectively, when compared against CNNs where $\#P_{target} = 10^6$. While the training time is understood to be lower for DWNNs, a deeper issue arises from kernel designed using LGNs and LTNs. LGCNNs use deep logic gate tree convolution (Petersen et al., 2024). In this design, each kernel passes $(k * k * channels * b)$ inputs through a sequence of layers that decrease in width by $\lceil npl_{Li-1}/n \rceil$, where $npl_{Li-1}$ is the number of nodes on the previous layer and $n$ is the number of logic gate ($n = 2$) or look-up-table ($2 \leq n \leq 6$) inputs. Each convolutional layer can have 2-7 LGN or LTN layers depending on the dataset, encoding scheme, and DWNN properties, where the number of convolutional layers generally increases with model size. These detrimental scaling traits are exacerbated by the loss functions of LGNs and LTNs, which are noisy and show low gradients (Figure 4). *This result highlights the need for efficient kernel designs for DWNN CNNs.*

$\Delta S$ - *Gap in Model Robustness Properties*: While noise-robust DWNNs hold value in real-world applications where sensor anomalies shift observations away from training distributions, study results indicate that DWNNs are generally less robust than their DNN counterparts and realize fewer robustness traits when scaling. Figure 4 compares the decay rate of model accuracy when injecting S&P noise and occlusion of increasing degree (see Appendix B). Generally, MLP and CNN models can learn to obtain robustness against S&P and occlusion noise (e.g. smaller $\beta$) at scale, though LGNs and LTNs are often more robust on simple datasets for small parameter counts.

---

[3]The relatively low accuracy on CIFAR10 results from using sparse CNN architectures while excluding batch normalization and dropout to ensure architectural consistency to the LGCNN and LTCNN architectures

Table 2: Test accuracy of the models in $P_{\Delta S}$. The top, middle, and bottom values in each cell represent the maximum, mean, and standard deviation of the test accuracy, respectively. Color scales are unique per dataset to highlight the mean performance. The second row represents $\#P_{target}$

| Model | MNIST | | | | | FASHIONMNIST | | | | | CIFAR10 | | | | |
|---|---|---|---|---|---|---|---|---|---|---|---|---|---|---|---|
| | $10^{4.0}$ | $10^{4.5}$ | $10^{5.0}$ | $10^{5.5}$ | $10^{6.0}$ | $10^{4.0}$ | $10^{4.5}$ | $10^{5.0}$ | $10^{5.5}$ | $10^{6.0}$ | $10^{4.0}$ | $10^{4.5}$ | $10^{5.0}$ | $10^{5.5}$ | $10^{6.0}$ |
| **MLP** | 90.5%
66.7%
±22.6% | 95.0%
92.9%
±1.4% | 96.9%
96.1%
±0.4% | 97.8%
97.4%
±0.3% | 98.0%
97.4%
±0.5% | 83.0%
66.1%
±20.5% | 84.8%
83.3%
±1.0% | 85.9%
85.2%
±0.4% | 87.1%
86.4%
±0.4% | 87.1%
86.3%
±0.7% | 26.1%
16.9%
±5.0% | 33.4%
24.0%
±4.6% | 39.7%
34.9%
±2.3% | 43.1%
40.2%
±1.9% | 45.3%
42.2%
±2.0% |
| **LGN** | 72.2%
69.7%
±1.2% | 83.9%
80.8%
±1.5% | 90.7%
87.2%
±1.7% | 95.9%
93.7%
±3.7% | 96.8%
75.6%
±35.7% | 70.5%
68.7%
±0.7% | 78.3%
76.3%
±0.9% | 82.8%
81.7%
±0.7% | 85.2%
83.7%
±2.8% | 85.3%
68.3%
±29.2% | 32.3%
31.2%
±0.5% | 35.8%
34.9%
±0.5% | 38.4%
37.4%
±0.6% | 41.0%
38.6%
±6.8% | 41.4%
31.2%
±11.8% |
| **LTN** | 85.6%
83.1%
±2.0% | 92.2%
90.3%
±1.2% | 96.4%
94.6%
±1.4% | 97.0%
95.8%
±0.9% | 96.9%
92.4%
±15.8% | 79.7%
77.5%
±1.3% | 83.1%
82.2%
±0.8% | 85.2%
84.3%
±0.6% | 85.4%
84.3%
±0.9% | 85.7%
80.7%
±13.7% | 37.4%
35.0%
±1.0% | 40.2%
38.8%
±0.7% | 41.2%
39.9%
±0.7% | 42.2%
38.6%
±2.0% | 39.6%
32.7%
±7.6% |
| **CNN** | 98.6%
88.7%
±18.4% | 98.8%
94.2%
±17.0% | 99.0%
93.2%
±18.1% | 99.3%
98.7%
±0.3% | 98.7%
87.7%
±28.2% | 87.4%
78.8%
±16.6% | 88.8%
83.7%
±15.1% | 89.6%
86.0%
±5.9% | 89.7%
88.2%
±0.4% | 88.9%
78.3%
±25.3% | 57.3%
40.6%
±11.0% | 59.6%
45.8%
±9.6% | 60.4%
48.8%
±8.1% | 62.4%
55.7%
±4.3% | 54.8%
43.8%
±15.0% |
| **LGCNN** | 75.8%
70.0%
±8.1% | 86.9%
75.7%
±17.6% | 93.1%
81.4%
±19.6% | 94.9%
76.9%
±23.4% | 97.2%
85.0%
±22.9% | 72.0%
63.9%
±7.6% | 79.2%
69.0%
±15.3% | 83.0%
74.8%
±14.0% | 86.3%
71.4%
±17.5% | 87.4%
74.1%
±19.9% | 31.2%
27.8%
±2.8% | 40.8%
31.6%
±6.9% | 45.6%
36.6%
±8.1% | 46.9%
33.5%
±9.3% | 44.6%
35.8%
±7.4% |
| **LTCNN** | 86.5%
77.7%
±14.8% | 94.0%
76.6%
±21.1% | 95.0%
88.5%
±11.5% | 97.2%
82.5%
±24.0% | 97.8%
85.1%
±22.8% | 80.3%
71.9%
±14.0% | 84.0%
71.4%
±13.5% | 85.7%
79.8%
±10.2% | 87.4%
75.4%
±17.3% | 87.7%
76.4%
±15.3% | 43.2%
34.2%
±7.1% | 47.4%
36.5%
±6.9% | 50.4%
41.6%
±8.1% | 52.2%
41.1%
±11.8% | 48.9%
35.3%
±11.6% |
| $\Delta \max_{DNN}$
$\Delta \mu_{DNN}$ | 8.1%
22.0% | 3.7%
1.3% | 2.1%
-2.9% | 1.5%
1.2% | 0.7%
-9.7% | 4.4%
12.6% | 4.1%
0.3% | 3.7%
0.7% | 2.6%
1.7% | 1.8%
-8.0% | 31.2%
24.2% | 26.2%
21.5% | 20.7%
14.0% | 19.3%
15.5% | 9.5%
1.6% |
| $\Delta \max_{LGN}$
$\Delta \mu_{LGN}$ | 26.4%
19.0% | 14.9%
13.5% | 8.3%
6.0% | 3.4%
4.9% | 1.9%
12.1% | 16.9%
10.1% | 10.5%
7.3% | 6.8%
4.2% | 4.5%
4.5% | 3.6%
10.0% | 25.0%
9.4% | 22.0%
10.9% | 22.0%
11.5% | 21.4%
17.1% | 13.4%
12.6% |
| $\Delta \max_{LTN}$
$\Delta \mu_{LTN}$ | 13.0%
5.6% | 6.5%
3.9% | 2.6%
-1.4% | 2.3%
2.9% | 1.8%
-4.8% | 7.7%
1.3% | 5.7%
1.5% | 4.5%
1.7% | 4.2%
3.9% | 3.2%
-2.4% | 19.9%
5.6% | 19.4%
7.0% | 19.1%
8.9% | 20.2%
17.1% | 15.2%
11.1% |

Table 3: Training time of the models in $P_{\Delta S}$. The top, middle, and bottom values in each cell represent the maximum, mean, and standard deviation of the training time, respectively. Color scales are unique per dataset to highlight the mean times. The second row represents $\#P_{target}$

| Model | MNIST | | | | | FASHIONMNIST | | | | | CIFAR10 | | | | |
|---|---|---|---|---|---|---|---|---|---|---|---|---|---|---|---|
| | $10^{4.0}$ | $10^{4.5}$ | $10^{5.0}$ | $10^{5.5}$ | $10^{6.0}$ | $10^{4.0}$ | $10^{4.5}$ | $10^{5.0}$ | $10^{5.5}$ | $10^{6.0}$ | $10^{4.0}$ | $10^{4.5}$ | $10^{5.0}$ | $10^{5.5}$ | $10^{6.0}$ |
| **MLP** | 4.3
3.4
±0.4 | 4.2
3.6
±0.3 | 4.3
3.8
±0.4 | 6.9
4.7
±1.1 | 8.4
4.1
±1.6 | 4.3
3.5
±0.4 | 4.2
3.6
±0.3 | 4.5
3.8
±0.4 | 6.9
4.7
±1.1 | 7.9
4.1
±1.6 | 4.4
3.7
±0.3 | 4.8
3.8
±0.3 | 4.6
3.9
±0.3 | 5.5
4.5
±0.6 | 9.6
6.6
±1.9 |
| **LGN** | 3.7
3.3
±0.2 | 3.9
3.2
±0.3 | 3.7
3.3
±0.2 | 5.0
3.7
±0.4 | 6.7
4.9
±0.6 | 3.8
3.3
±0.2 | 3.9
3.3
±0.3 | 3.7
3.3
±0.2 | 5.0
3.8
±0.4 | 7.0
5.0
±0.6 | 4.2
3.6
±0.3 | 4.1
3.5
±0.3 | 4.1
3.5
±0.2 | 4.9
3.9
±0.4 | 6.8
4.9
±0.6 |
| **LTN** | 3.7
3.2
±0.2 | 3.6
3.3
±0.2 | 4.2
3.5
±0.3 | 5.3
4.0
±0.6 | 5.3
4.5
±0.4 | 3.7
3.2
±0.2 | 3.7
3.3
±0.2 | 4.3
3.6
±0.2 | 5.4
4.0
±0.6 | 5.2
4.5
±0.4 | 3.9
3.5
±0.2 | 4.0
3.5
±0.3 | 4.6
3.8
±0.4 | 5.2
4.1
±0.5 | 5.4
4.5
±0.4 |
| **CNN** | 6.3
3.9
±0.8 | 4.7
3.7
±0.4 | 4.2
3.4
±0.3 | 5.0
3.7
±0.4 | 5.1
3.8
±0.6 | 6.4
3.9
±0.8 | 4.5
3.6
±0.3 | 4.1
3.4
±0.3 | 4.9
3.6
±0.4 | 5.3
3.7
±0.6 | 7.4
4.3
±1.0 | 5.0
3.9
±0.4 | 4.6
3.7
±0.4 | 5.0
3.9
±0.3 | 5.3
4.1
±0.6 |
| **LGCNN** | 12.7
7.1
±2.1 | 33.3
12.5
±8.9 | 80.9
32.2
±20.1 | 132.0
39.0
±33.0 | 88.1
30.9
±29.5 | 13.1
7.2
±2.2 | 33.4
12.7
±9.1 | 81.6
32.5
±20.7 | 140.8
39.9
±36.2 | 87.7
31.6
±32.8 | 12.7
7.5
±2.0 | 36.1
13.7
±10.3 | 78.7
33.5
±20.0 | 121.3
38.7
±33.9 | 85.8
28.9
±29.0 |
| **LTCNN** | 119.1
8.0
±15.2 | 82.1
17.7
±17.9 | 100.1
11.8
±17.0 | 58.0
12.9
±9.9 | 55.7
12.0
±11.4 | 118.9
7.8
±15.1 | 71.1
15.0
±13.4 | 107.0
10.5
±16.2 | 64.3
12.5
±9.9 | 43.3
11.1
±9.3 | 99.8
8.2
±12.7 | 82.0
15.4
±12.3 | 116.5
10.8
±17.1 | 52.5
12.6
±8.3 | 36.6
10.6
±7.9 |

However, LGN, LTN, LGCNN, and LTCNN architectures tend to become more brittle (e.g. $\beta$ increases or remains constant) when model size increases. Note one exception is observed for LGNs and LTNs, where models become more robust to occlusions for FASHIONMNIST. As these results represent models trained without data augmentation, it is noteworthy that robustness improvves for MLPs and CNNs but declines for LGN and LTN-based models at scale. This observation confirms established findings by Aleksander et al. (2009) which observe that generalization in WNNs tends to decline with increase model size. Despite the value in letting DWNNs learn using backpropagation, this trait fails to achieve scaling-robustness properties of DNNs. While methods like "bleaching" exist to mitigate such issues (Susskind et al., 2022), *continued research could improve model architectures and training strategies for discrete manifolds. While current research mimics DNN architectures (e.g. LGCNN & LTCNN), focusing future work on developing noise-aware learning mechanisms may fundamentally improve network robustness.*

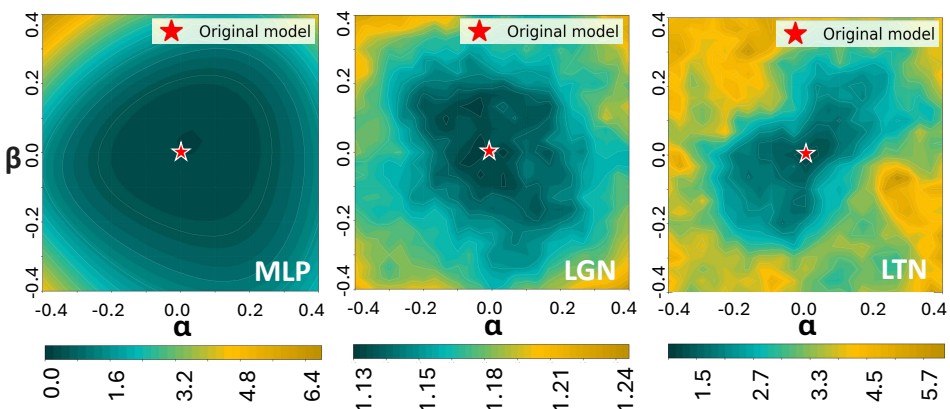

Figure 3: MLP, LGN, and LTN loss surfaces $\#P_{target} = 10^5$, following work from Li et al. (2018)

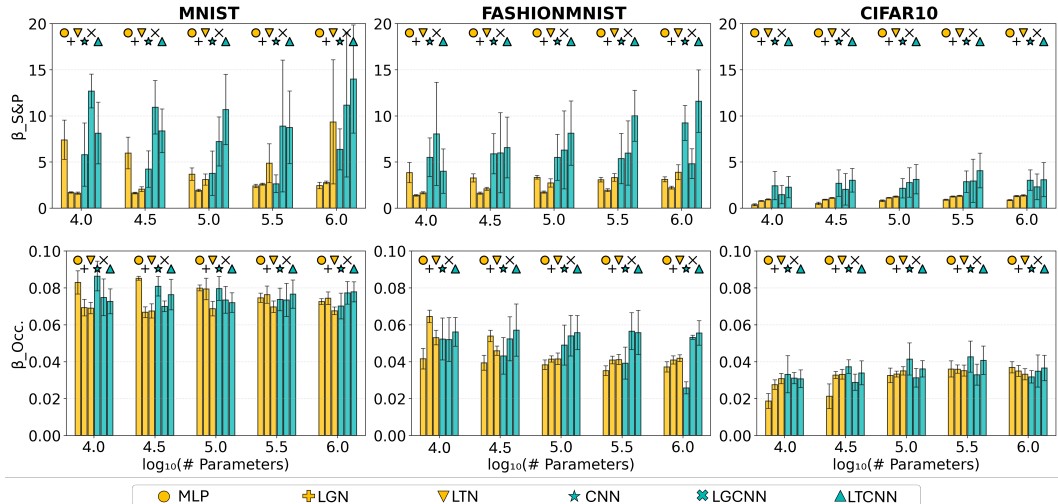

Figure 4: Decay rate of models' test accuracy when injecting S&P (top) and occlusions (bottom)

$\Delta D$ - *Gap in Augmentation-Based Generalization*: Data augmentation generally increases a model's learned robustness by altering the training distribution to overlap with noisy samples, though the $\Delta D$ study reveals a gap in the robustness gains realized for DWNNs. Table 4 shows baseline accuracy and robustness measures (i.e. $\%A_{Base}$, $\beta_{Occ.Base}$, $\beta_{S\&PBase}$) alongside changes realized when training with data augmentation (i.e. $\Delta\%A$, $\Delta\beta_{Occ.}$, $\Delta\beta_{S\&P}$). Here, data augmentation generally has a minimal impact on test accuracy for the studied size range (e.g. -1.0% ± 1.4% across all variations) though shows varied impacts for robustness. Augmentation generally improves robustness for MLPs, letting them surpass the performance of LGNs and LTNs across all size scales and datasets excluding an anomaly on CIFAR10 when $\#P_{target} = 10^{5.0}$. As MLPs generally benefit the most from data augmentation and achieve the best robustness, it remains noteworthy that LGNs retain the best S&P robustness both prior to and after data augmentation. Prior work developed specialized salience maps for explaining LGNs, showing that LGNs generally learn to look for input patterns which represent the class average for MNIST and FASHIONMNIST (Wormald et al., 2025) rather than memorizing noisy samples. This observation may explain why LGNs are robust to S&P and sensitive to occlusions - while S&P alters individual pixels that less frequently corrupt important image regions, occlusions are more likely to cover sensitive regions learned from the class average. *Continued research could target the inefficiency gap in training DWNNs with data augmentation. Comparing these results with prior findings may reveal an opportunity to use salience maps as an indicator for brittle features and thereby inform training to improve robustness.*

Table 4: Testing accuracy and model robustness to occlusions and S&P noise for two target sizes. "Base" indicates a parameter baseline when trained without augmentation, when "$\Delta$" indicates the change in a given metric when the same models are trained using data augmentation

| Metric | $\#P_{target}$ | MNIST | | | FASHIONMNIST | | | CIFAR10 | | |
|---|---|---|---|---|---|---|---|---|---|---|
| | | MLP | LGN | LTN | MLP | LGN | LTN | MLP | LGN | LTN |
| $\%A_{Base}$ | $10^{4.5}$ | 93.5% | 80.8% | 90.3% | 83.7% | 76.3% | 82.2% | 24.0% | 34.9% | 38.8% |
| $\Delta\%A$ | $10^{4.5}$ | -4.5% | -2.0% | -1.0% | -0.4% | -1.6% | -2.6% | +0.7% | +0.6% | -0.3% |
| $\%A_{Base}$ | $10^{5.0}$ | 96.3% | 87.2% | 94.9% | 85.4% | 81.7% | 84.4% | 34.9% | 37.4% | 39.9% |
| $\Delta\%A$ | $10^{5.0}$ | -1.0% | -1.6% | -2.5% | -0.3% | -1.7% | -0.3% | -1.3% | -0.2% | +1.2% |
| $\beta_{\text{Occ. Base}}$ | $10^{4.5}$ | 0.080 | 0.068 | 0.069 | 0.037 | 0.056 | 0.043 | 0.082 | 0.035 | 0.031 |
| $\Delta\beta_{\text{Occ.}}$ | $10^{4.5}$ | -0.027 | +0.001 | -0.000 | -0.009 | -0.002 | -0.002 | -0.064 | -0.005 | -0.005 |
| $\beta_{\text{Occ. Base}}$ | $10^{5.0}$ | 0.077 | 0.069 | 0.070 | 0.037 | 0.045 | 0.040 | 0.025 | 0.035 | +0.036 |
| $\Delta\beta_{\text{Occ.}}$ | $10^{5.0}$ | -0.028 | -0.001 | -0.004 | -0.008 | -0.002 | -0.003 | +0.053 | -0.006 | -0.008 |
| $\beta_{\text{S\&P Base}}$ | $10^{4.5}$ | 4.073 | 1.766 | 3.259 | 3.186 | 1.661 | 2.671 | 3.937 | 0.927 | 1.091 |
| $\Delta\beta_{\text{S\&P}}$ | $10^{4.5}$ | -2.415 | -0.456 | -2.046 | -1.842 | -0.574 | -1.470 | -3.408 | -0.092 | -0.238 |
| $\beta_{\text{S\&P Base}}$ | $10^{5.0}$ | 2.879 | 2.122 | 4.768 | 3.150 | 1.941 | 3.168 | 0.727 | 1.122 | 1.293 |
| $\Delta\beta_{\text{S\&P}}$ | $10^{5.0}$ | -1.639 | -0.924 | -3.344 | -1.848 | -0.809 | -1.868 | +6.682 | -0.214 | -0.324 |

*$\Delta B \& \Delta E$ - Gap in $b$-Sensitivity under Varied Encodings*: While LGNs and LTNs derive less benefit from data augmentation than MLPs, they can be more sensitive to encoding schemes and bit depth. Figure 4 illustrates the improvement in model accuracy with increasing bit depth, expressed as the $b$-sensitivity ($\Delta\%A/\Delta b$; Appendix D), for both quantization and thermometer encodings. Our findings show that thermometer encodings often improve $\Delta\%A/\Delta b$, which agrees with prior work (Carneiro et al., 2015). However, the magnitude of improvement varies per model type. Quantization improves accuracy only in certain regimes (e.g., $b$-sensitivity $> 0$ when $\#P_{target} > 10^{4.25}$ for FASHIONMNIST), and tends to perform worse on average, though its disadvantages diminish at larger model scales. While thermometer encodings provide consistent performance gains, they introduce substantial bandwidth costs: for example, a 256×256 RGB image requires 196.6kB in 8-bit quantized form but 6.27MB using an uncompressed thermometer encoding. Even for smaller datasets like CIFAR10 (32×32 RGB pixels), the required storage for thermometer encodings grows from 3.1 kB/image (quantization) to 97.9 kB/image. These comparisons highlight a gap between the practical data-handling efficiency of DNNs and DWNNs. While DWNNs have achieved near-SOTA performance on CIFAR10 with b = 5 (~1.9 kB/image), it is unclear whether such levels of compression will scale to larger datasets with more classes, especially given the increasing representational demands of real-world tasks. To lessen these gaps, *future works may explore embedding layers (e.g. n-input LTNs could adaptively learn encodings per pixel, where each pixel represents an n-bit quantized value) and methods which improve performance while using smaller bit depths.*

Aside from the gap in data efficiency, some attention may be drawn to DWNN performance on CIFAR10, where LGNs and LTNs generally outperform their MLP counterparts regardless of bit depth and encoding scheme. Hypothetically, MLPs may be expected to leverage the information from larger $b$ more effectively as each neuron holds the full input in its receptive field. Conversely, each gate in LGNs and LTNs have significantly lower receptive fields (e.g. the state of a logic gate two layers into an LGN depends only on 4 inputs). When pixels are encoded to multiple bits (i.e. $b > 1$), information from each pixel may be split between independent sub networks in LGNs and LTNs. It may be expected that splitting information between sub-networks and reducing a network's receptive field would reduce model performance for more complex datasets, though the opposite is observed. These benefits may highlight *value in researching ensembles of small DWNNs which independently perform well on subsets of classes. While ensemble methods exist for classical RAM-based WNNs (Lusquino Filho, 2021), systematic studies on DWNN ensembles remain limited.*

*$\Delta M$ - Gap in Model Performance when Using Learnable Mappings*: Learnable mappings improve model performance by enabling adaptive inter-layer connections in LGN and LTN architectures, but can increase parameter counts and training times by an order of magnitude. Table 5 summarizes the effects of learnable mappings across datasets, highlighting changes in test accuracy, training

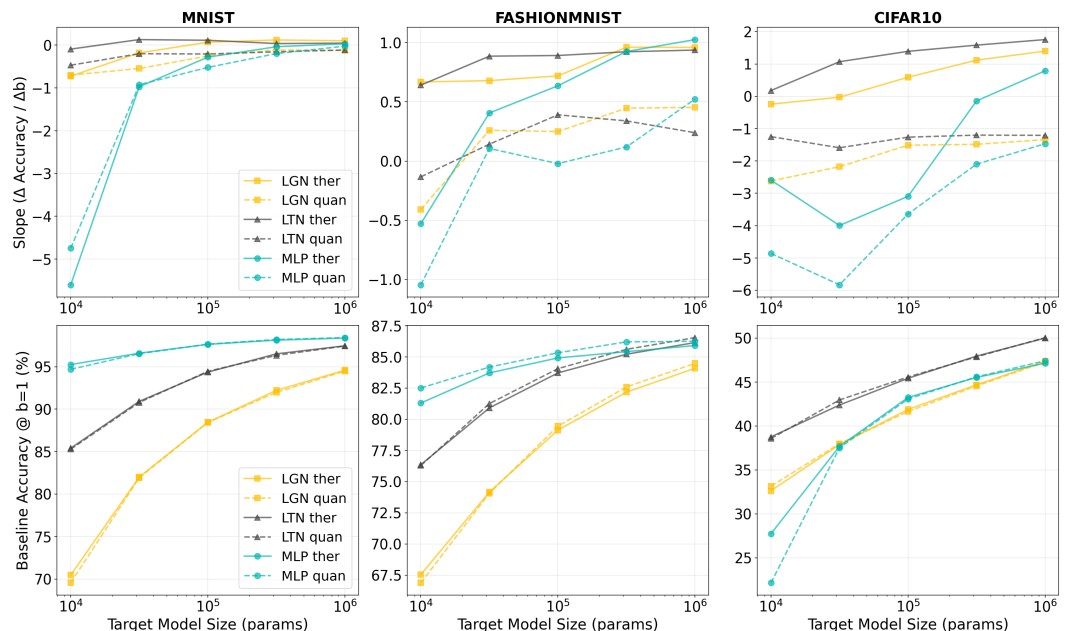

Figure 5: Mean test accuracy of LGNs, LTNs, and MLPs trained with varied bit depths across a range of target sizes. The top row shows the bit depth sensitivity, where the bottom row show the baseline accuracy when $b$=1

time, and model size. The impact on test accuracy depends on whether baseline models are close to saturating model performance for a given dataset. Test accuracy increases across most model configurations (6-20% depending on dataset and architecture), where LGNs benefit more consistently than LTNs. Note the distribution of $\Delta\%A$ for LTNs sometimes reduces the accuracy beneath baseline performance (FASHIONMNIST and CIFAR10 when $\#P_{target} = 10^{4.5}$) which is not observed for LGNs. The marginal benefit of learnable mappings may be lower for LTNs as their baseline accuracy is higher, perhaps because $n \geq 2$, so networks with randomly initialized interconnects have a higher chance of routing information to improve performance compared to LGNs. However, increasing $n$ is costly. Learnable mapping introduce $(npl_i * npl_{i-1} * n)$ additional parameters per layer, so the number of parameters tends to increase significantly more for LTNs. This growth in parameters translates to longer training times, where smaller models experience a $\sim$1.25X increase and larger models may require up to 3X more time (e.g., LGN CIFAR10, size $10^{4.5}$). LGNs achieve larger gains with smaller training-time penalties as $n = 2$, though the final accuracy tends to be lower on average. Despite the performance improvement, there is a significant gap between the number of parameters needed by DNNs and DWNNs to achieve comparable model accuracy on simple datasets. MLPs can obtain comparable accuracies for MNIST and FASHIONMNIST while using 2.9-48.4% of the parameters count ($\%\#P^4$ in Table 5), though require 67-337% the number of parameters to achieve comparable accuracies for CIFAR10. These observations show that learnable mappings may improve or hinder the "parameter efficiency" (or the number of parameters needed to reach a baseline performance) for certain problems, where there is a higher advantage for complex datasets. Considering the complexity of real-world datasets, *future research may explore model architectures and training strategies which improve the parameter efficiency of learnable mappings to simultaneously improving model accuracy while minimizing training times for complex datasets.*

## 5    CONCLUSIONS

This work provides the first comprehensive benchmark comparing differentiable Logic Gate Networks (LGNs) and Look-Up Table Networks (LTNs) against traditional MLPs and CNNs across

---

[4]$\%(\#P)$ is calculated as $(\#P_{MLP}/\#P_{LGN,LTN})$ where $\#P_{MLP}$ is the number of parameters needed for an MLP to achieve the test accuracy of DWNNs trained using learnable mappings, as determined using linear interpolation and extrapolation from entries in Table 2. $\#P_{LGN}$ and $\#P_{LTN}$ are determined from Table 5

Table 5: Impact of learnable mappings on test accuracy, training time, and the number of parameters. "B." indicates baseline values, "$\Delta$" represents how the baseline changed when using learnable mappings. "M", "F", and "C" represent MNIST, FASHIONMNIST, and CIFAR10, respectively

| | Model | Size | Accuracy (%) | | Time (s) | | $\text{Log}_{10}$(# Parameters) | | |
|---|---|---|---|---|---|---|---|---|---|
| | | | B. | $\Delta$ | B. | $\Delta$ | B. | $\Delta$ | %(#P) |
| M | LGN | $10^{4.0}$ | 69.7 | +(19.7±1.3) | 3.28 | +(2.1±0.6) | 4.1 | +(5.3±0.00) | 13.3% |
| | LGN | $10^{4.5}$ | 81.2 | +(13.8±2.2) | 3.26 | +(4.5±2.0) | 4.5 | +(5.3±0.0) | 34.7% |
| | LTN | $10^{4.0}$ | 83.1 | +(12.5±1.9) | 3.24 | +(2.3±0.7) | 4.2 | +(6.7±0.0) | 4.2% |
| | LTN | $10^{4.5}$ | 91.1 | +(4.3±5.0) | 3.33 | +(11.9±10.9) | 4.7 | +(6.8±0.2) | 3.9% |
| F | LGN | $10^{4.0}$ | 68.7 | +(12.0±0.8) | 3.30 | +(2.2±0.6) | 4.1 | +(5.3±0.0) | 11.8% |
| | LGN | $10^{4.5}$ | 76.6 | +(8.4±1.2) | 3.28 | +(4.4±2.0) | 4.5 | +(5.3±0.0) | 48.4% |
| | LTN | $10^{4.0}$ | 77.5 | +(7.9±1.4) | 3.26 | +(2.4±0.7) | 4.2 | +(6.7±0.0) | 5.4% |
| | LTN | $10^{4.5}$ | 82.6 | +(1.8±3.8) | 3.37 | +(11.9±11.0) | 4.7 | +(6.8±0.2) | 2.9% |
| C | LGN | $10^{4.0}$ | 31.3 | +(9.6±0.6) | 3.56 | +(5.8±0.6) | 4.1 | +(5.9±0.0) | 67.8% |
| | LGN | $10^{4.5}$ | 35.1 | +(9.3±1.2) | 3.53 | +(12.6±1.3) | 4.5 | +(5.8±0.0) | 337.4% |
| | LTN | $10^{4.0}$ | 35.0 | +(9.6±1.3) | 3.48 | +(11.8±1.4) | 4.2 | +(6.3±0.1) | 159.2% |
| | LTN | $10^{4.5}$ | 39.0 | +(4.5±5.6) | 3.62 | +(22.8±9.7) | 4.7 | +(6.1±0.2) | 143.7% |

axes of scalability, robustness, and training efficiency. By introducing the LTCNN architecture and the open-source wn$^2$Architect library, we confirm that convolutional DWNNs can achieve test accuracies competitive with standard DNNs on small-scale datasets. However, our systematic evaluation across 4,350 model variations quantifies performance gaps that hinder the scalability of DWNNs to real-world applications. Key findings are summarized below:

*The Scalability Gap:* While the proposed LTCNN reduces training time by 2-4X compared to LGCNNs, both architectures scale poorly compared to standard CNNs. Large-scale DWNNs suffer from a 2.8-10X training time penalty and diminishing returns in accuracy, largely due to the depth of logic-gate kernels and inefficient gradient propagation (Table 2).

*The Robustness Gap:* Contrary to the behavior of MLPs and CNNs, increasing the scale of DWNNs does not generally improve robustness. Our results confirm that scaled-up LGNs and LTNs tend to become more brittle rather than learning generalized features. While LGNs demonstrate superior intrinsic resistance to Salt-and-Pepper noise, they struggle significantly with occlusions (Figure 4).

*The Bandwidth Gap:* Findings confirm that thermometer encodings improve performance (e.g. $\Delta\%A/\Delta b$,) but this scheme imposes a memory bandwidth tax (increasing data size by $\sim$32X for 8-bit images). Quantization avoids this cost but yields sub-optimal accuracy (Figure 4).

*Parameter Efficiency Gap:* While learnable DWNN interconnects consistently improve maximum accuracy (by 6-20%), they increase computational costs significantly (Table 5).

Future work should target the specific gaps in training efficiency, robustness, and parameter efficiency quantified in this study. While developing optimized CUDA kernels for bit-wise operations remains a practical necessity to replace heavy floating-point simulations (Petersen et al., 2024), fundamentally improving gradient stability requires architectural innovations, such as reducing the depth of logic-gate kernels for LGCNNs and LTCNNs. Furthermore, to address the observed brittleness in scaled models, the field must move beyond mimicking CNN architectures and instead develop regularization techniques tailored to discrete manifolds. Promising research directions include: (1) investigating sparse initialization strategies to mitigate the computational costs of learnable interconnects; (2) replacing bandwidth-heavy thermometer encodings with learnable, adaptive embedding layers which learn pixel-specific encodings; and (3) exploring how network saliency maps could expose brittle feature to inform network training methods and improve robustness.

By resolving these fundamental gaps in training dynamics and representation, the field can move closer to realizing the promise of DWNNs: deep, robust, and interpretable logic networks capable of operating efficiently in resource-constrained environments.

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

## A  DETAILS ON THE MODEL ARCHITECTURE SEARCH

Architecture variations for $P_{\Delta S}$ were selected by parameterizing each model type and using an elitist evolutionary algorithm for neural-architecture search targeting selected model sizes. Figure 2 visualizes the high-level network architectures. The feed forward networks vary the number of layers ($L_f$) and the nodes per layer ($npl$), where $npl_0$ (the first layer) is permitted to decrease by a factor of $m_{npl}$ per layer (i.e. $npl_{i+1} = \lceil npl_i * m_{npl} \rceil$). Convolutional architectures use a set of convolution blocks followed by a feed-forward network. The convolution block constrains the number of convolutional layers ($L_c$), the number of kernels ($N_k$) per $L_c$, and the kernel size, $k$. While $k$ is held constant across all kernels per architecture, $N_k$ is increased per $L_c$ according to $N_{k,i+1} = \lceil N_{k,i} * m_{N_k} \rceil$ where $i$ is the current layer, $N_{k,0}$ is selected by the search algorithm as $N_k$, and $m_{N_k}$ is the kernel increase factor. The downstream forward network sets the initial $npl_0$ equal to the number of output features, where $m_{npl}$ adjusts the follower layers as before. Here, the $npl$ on the final layer is constrained to be divisible by 10 for LGN and LTN networks. Outputs are passed into GroupSum and Argmax operations for the LGCNN and LTCNN networks, and to 10 output nodes for the CNN. Ranges for these parameters are defined to constrain the search space $S$ as defined in Table 1, where another parameter $n$ defines the number of inputs per LTN, which is held constant per LTN-based variant. Note the original LGCNN paper constrains the number of input channels per kernel ($Q$) to reduce the model size and improve training efficiency, which is applied to the CNN, LGCNN and LTCNN networks and selected from the set {-, 1, 2, 4, 8}, where "-" indicates all channels were used. Finally, each kernel in the LGCNN and LTCNN networks are constructed as complete n-ary trees, or complete binary trees for LGCNN kernels, where inputs are randomly mapped to the receptive field upon initialization.

Algorithm 1 performs a constrained elitist guided random search over the parameter space $S$ to produce architectures $P_{scale} = \{A_1, A_2, ..., A_n\}$ where predicted parameter counts $\#P = size(A_n)$ are within a tolerance $\gamma$ of the size target $\#P_{target}$. Each iteration draws a candidate either by uniform random sampling or by mutating an elite from the pool $E$ using a mutation operator $M$ with a linearly decaying noise $\sigma(...)$. If the error $e = |size(A_n) - \#P_{target}|$ is less than $\gamma$, the candidate is added to the solution set $\Omega$. An elite pool $E$ of up to $B$ candidates is maintained with an associated set of errors $\epsilon$ which guides future sampling. When the pool is not full, new candidates are appended, otherwise the worst elite is replaced if a newer candidate yields a smaller error. Sampling balances exploration and exploitation via a fixed probability $p = 0.25$ to sample uniformly or via elitist mutation with parent selection weighted by $exp(-\epsilon/\alpha)$. Feasibility is enforced through explicit constraint checks and grid snapping for discrete parameters to ensure the number of outputs is divisible by ten. The resulting distribution of parameters is displayed in Figure A.

**Algorithm 1** Size-constrained guided random search with elitist selection

**Require:** $\#P_{target}$, $\gamma$, $S$, feasible set $C \subseteq S$, $size(A_n) : A_n \to \mathbb{N}$, $p \in [0, 1]$, max iterations $M$, $B$, $\sigma$, mutation operator $\mathcal{M}(A_n; \sigma)$, projection $\Pi_C : S \to C$

**Ensure:** set $\Omega \subseteq C$ with $|s(c) - \#P_{target}| \le \gamma$

1: $E \leftarrow \emptyset$, $\varepsilon \leftarrow \emptyset$, $\Omega \leftarrow \emptyset$
2: **for** $t = 0$ to $M - 1$ **do**
3:      **if** $|\Omega| = N$ **then**
4:          **break**
5:      **end if**
6:      **if** $E = \emptyset$ or Bernoulli$(p) = 1$ **then**
7:          $c \leftarrow \Pi_C\big(\text{Uniform}(S)\big)$
8:      **else**
9:          $\alpha \leftarrow \max\{1, \text{median}(\varepsilon), \#P_{target}/\max(2, \#P_{target})\}$
10:        choose $i$ with $\mathbb{P}(i) \propto \exp(-\varepsilon_i/\alpha)$; $P \leftarrow E_i$
11:        $c \leftarrow \Pi_C\big(\mathcal{M}(P; \sigma(t))\big)$
12:      **end if**
13:     $e \leftarrow |s(c) - \#P_{target}|$
14:     **if** $e \le \tau$ **then**
15:        $\Omega \leftarrow \Omega \cup \{c\}$
16:     **end if**
17:     **if** $|E| < B$ **then**
18:        append $c$ to $E$; append $e$ to $\varepsilon$
19:     **else**
20:        $j \leftarrow \arg\max_i \varepsilon_i$
21:        **if** $e < \varepsilon_j$ **then**
22:          $E_j \leftarrow c$; $\varepsilon_j \leftarrow e$
23:        **end if**
24:     **end if**
25: **end for**
26: **return** $\Omega$

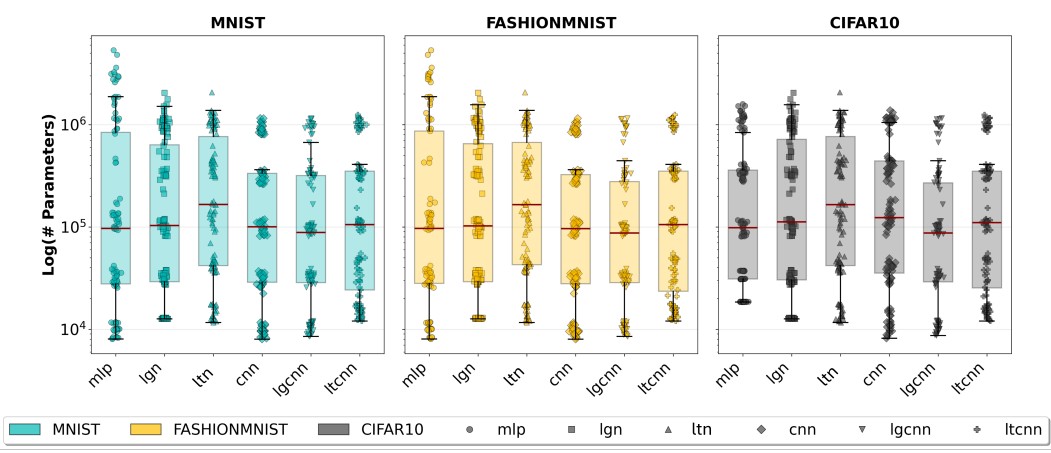

Figure 6: Distribution of parameter counts for model architectures generated by the evolutionary search algorithm

## B  QUANTIZATION AND THERMOMETER ENCODING

All images were uniformly quantized prior to training in every study except the $\Delta E$ experiment. Let $x$ denote an input pixel value and $b$ the number of quantization bits. The uniform quantizer is defined as

$$x_{\text{quant}} = \text{round}\left(\frac{x - x_{\min}}{x_{\max} - x_{\min}} \cdot (2^b - 1)\right),$$

where $x_{\min}$ and $x_{\max}$ are the minimum and maximum pixel values observed across the dataset. The resulting integer code $x_{\text{quant}} \in \{0, \ldots, 2^b - 1\}$ is subsequently converted to a binary vector representation and used as input for all model types.

For thermometer encoding with $b$ bits per pixel, we use $b$ thresholds fitted to the training data range. Given a training dataset with pixel values in $[x_{\min}, x_{\max}]$, we define uniformly-spaced thresholds as

$$\tau_i = x_{\min} + \frac{i}{b+1} \cdot (x_{\max} - x_{\min}), \quad i \in \{1, 2, \ldots, b\}.$$

For normalized images where $x_{\min} = 0$ and $x_{\max} = 1$, this simplifies to $\tau_i = \frac{i}{b+1}$ for $i \in \{1, \ldots, b\}$. The thermometer encoding $\mathbf{T}(x) \in \{0, 1\}^b$ is defined element-wise as

$$\mathbf{T}(x) = (1[x > \tau_1], \, 1[x > \tau_2], \, \ldots, \, 1[x > \tau_b]),$$

This produces a monotonic encoding where higher input values yield longer contiguous runs of ones. For example, with $b = 2$, we have 2 thresholds at $\{0.33, 0.67\}$, yielding a 2-bit code, where a pixel value $x = 0.5$ encodes as $(1, 0)$.

## C  DECAY RATE ANALYSIS

To characterize robustness, denoted as $\beta$, we introduce two forms of input perturbations: (i) rectangular occlusions with widths and heights ranging from 0 to 20 pixels, and (ii) salt-and-pepper (S&P) noise occupying between $0\%$ and $50\%$ of image pixels. Figure B illustrates the corresponding degradation in test accuracy for the top five models targeting the $10^5$ parameter scale. The top row shows accuracy decay under increasing degrees of S&P noise, while the bottom row shows accuracy decay as occlusion size increases. Each subplot corresponds to a different dataset, and each curve represents one of the evaluated model classes (MLP, LGN, LTN, CNN, LGCNN, LTCNN).

In both perturbation settings, accuracy decreases smoothly and approximately exponentially as noise severity increases. Following this empirical behavior, exponential functions of the form

$$y = Ae^{-\beta x} + C$$

are fit to the decaying accuracy curves for each model. The parameter $\beta$ quantifies the rate at which test accuracy deteriorates with increasing perturbation strength: smaller values of $\beta$ indicate higher robustness, while larger values correspond to rapid degradation. These $\beta$ values are then compared across model families and parameter counts to characterize how robustness varies with model size and architecture.

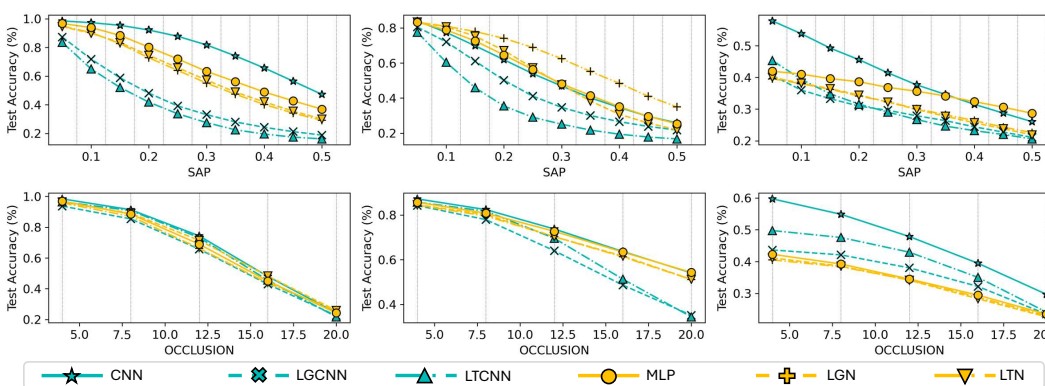

Figure 7: Degradation of test accuracy in response to occlusion noise and S&P noise (top and bottom rows, respectively) for the top 5 models targeting the size $10^5$, as ranked by test accuracy. The occlusion range represents the maximum x and y dimensions of a given occlusion, sampled randomly and independently. The occlusion is placed randomly in an image and replaces included values with zeros. The degree of S&P noise represents the percentage of pixels replaced with either salt or pepper noise, where masked pixels are randomly transitioned to salt or pepper noise with 50% probability

## D  BIT DEPTH SENSITIVITY ANALYSIS

To characterize bit-depth sensitivity, we evaluate how test accuracy changes as the bit depth ($b$) of the input encoding increases, where sensitivity is defined as $\Delta Accuracy / \Delta b$, representing the slope of the lines observed in Figure D. These trends illustrate how quantization (top rows) and thermometer encoding (bottom rows) alter model performance when increasing the model scale across MNIST, FASHIONMNIST, CIFAR10. Each subplot reports mean test accuracy over five model architectures. Here, the bit depth represents the number of bits used to encode pixel values in the respective encoding scheme (Appendix B), where larger values of $b$ make numerical values more precise. The trends in Figure D fit with a linear equation of the form $y = mx + c$, where results are summarized in Figure 4 from the main discussion.

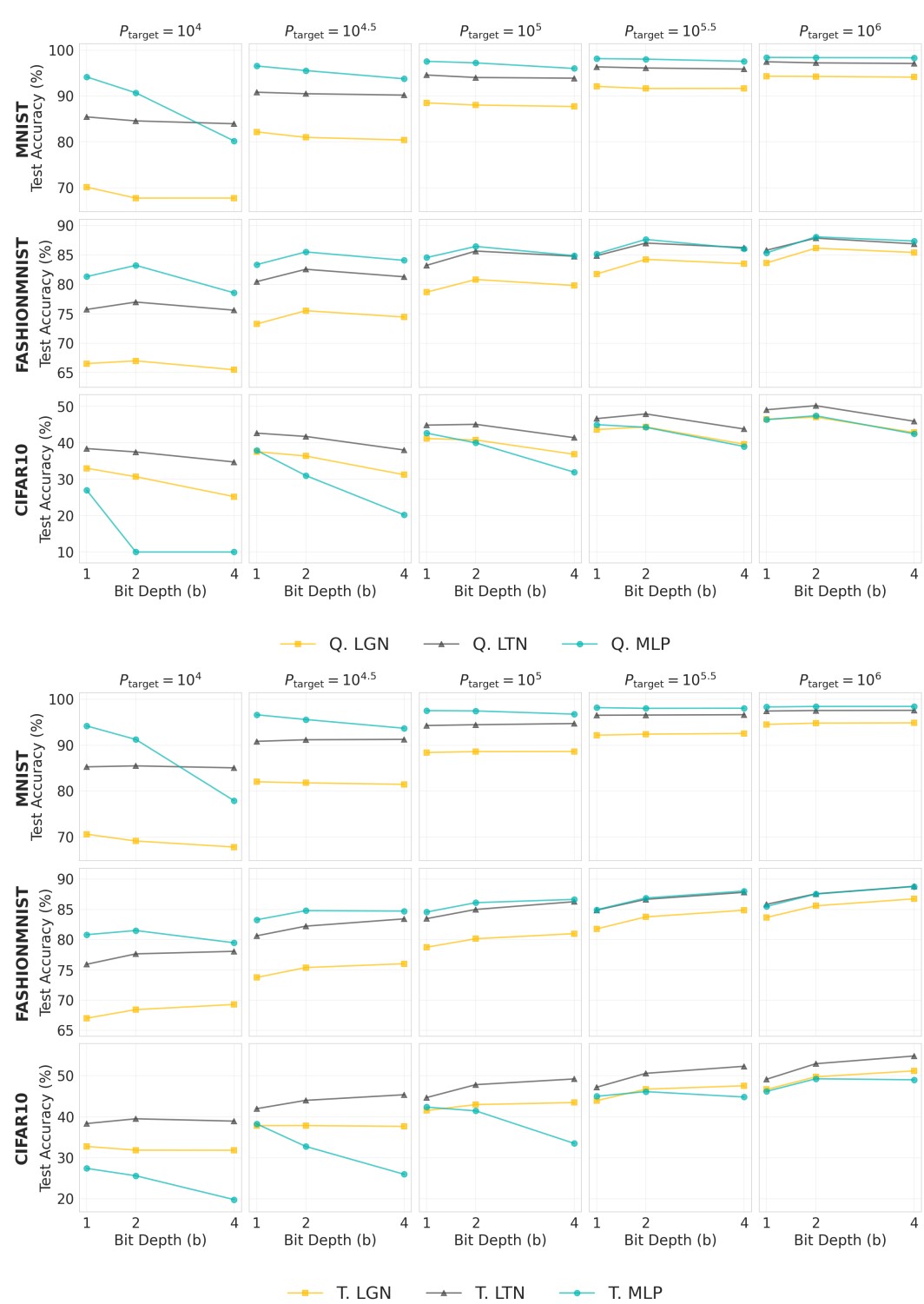

Figure 8: Test accuracy as a function of bit depth for quantization (top) and thermometer (bottom) encoding methods across three datasets (MNIST, FASHIONMNIST, CIFAR10) and five target model sizes ($\#P_{target} = 10^4$ to $10^6$ parameters). Each subplot shows results for MLPs, LGNs, and LTNs. Plots show the mean accuracies across five runs

# E  LGCNN AND LTCNN ARCHITECTURES

In this work, we benchmark two distinct classes of convolutional Differentiable Weightless Neural Networks (DWNNs): the Logic Gate Convolutional Network (LGCNN) and the Look-Up Table Convolutional Network (LTCNN). Both architectures replace the affine transformations (dot-products) characteristic of standard Deep Neural Networks (DNNs) with learnable, hierarchical Boolean function approximators. Formally, for an input patch $\mathbf{x} \in \mathbb{R}^N$ extracted from a local receptive field, both architectures define a mapping $\mathcal{K} : \mathbb{R}^N \to \mathbb{R}$ composed of differentiable logic units arranged in a tree topology. Table E summarizes the structural and mathematical differences between the two architectures. The primary distinction lies in the trade-off between node complexity and graph depth: LGCNNs utilize simple nodes in deep graphs, whereas LTCNNs utilize high-capacity nodes in shallow graphs.

Table 6: Mathematical Comparison of LGCNN and LTCNN Kernel Topologies

| Feature | LGCNN (Logic Gate Tree) | LTCNN (Look-Up Table Tree) |
|---|---|---|
| **Basis Function** | Expectation over 16 Boolean Operators | Continuous Relaxation of Truth Table |
| **Graph Topology** | Complete Binary Tree (Arity 2) | Complete $n$-ary Tree (Arity $n$) |
| **Kernel Depth** | $\mathcal{O}(\log_2 N)$ | $\mathcal{O}(\log_n N)$ |
| **#Parameters per Node** | 16 (Weights for Softmax) | $2^n$ (Table Entries) |

The LGCNN extends the LGN to the convolutional domain, as introduced by Petersen et al. (2024). The kernel is structured as a deep **binary tree** of learnable 2-input logic gates.

**LGCNN Mathematical Structure:** Let the receptive field of the kernel contain $N = k \times k \times C_{in}$ inputs. The LGCNN approximates the local function via a complete binary tree of depth $D \approx \lceil \log_2 N \rceil$. The leaves of the tree are the input features, and internal nodes are differentiable binary operators. Each node $f_i$ computes a function $f_i : [0,1]^2 \to [0,1]$. During training, the operation is relaxed probabilistically. Let $\mathcal{B}$ be the set of all 16 possible Boolean functions for two inputs. The output of a node is defined as the expectation over $\mathcal{B}$:

$$y = \sum_{j=1}^{16} p_j \cdot g_j(a, b) \tag{1}$$

where $a, b$ are inputs from child nodes, $g_j \in \mathcal{B}$, and $p_j$ is a learned probability mass derived from a softmax over learnable weights.

**LGCNN Complexity and Gradient Flow:** Because the tree is binary (arity $n = 2$), the path length from the root to any input leaf scales logarithmically with base 2. For high-dimensional inputs, this results in a deep computational graph, potentially subjecting the model to vanishing gradient phenomena typical of deep architectures, as the error signal must traverse $\log_2 N$ non-linear transformations to update the input layer weights.

The LTCNN generalizes the Boolean convolution introduced by Petersen et al. (2024) by replacing the binary tree with a hierarchical $n$-**ary tree** of differentiable Look-Up Tables (LUTs). This architecture leverages the higher capacity of $n$-input LUTs ($n \geq 2$) to construct shallower, more expressive kernels.

**LTCNN Mathematical Structure:** The LTCNN kernel is defined as a tree of differentiable LUTs, each with arity $n$. For a receptive field of size $N$, the topology is constructed recursively by reducing the dimensionality by a factor of $n$ at each layer. Consequently, the depth of the kernel is $D \approx \lceil \log_n N \rceil$. Each node is parameterized by a real-valued truth table $\mathbf{T} \in \mathbb{R}^{2^n}$. For binary inputs $\mathbf{x} \in \{0,1\}^n$, the node acts as a direct indexing operation. During training, inputs are continuous, and the lookup is relaxed via a multilinear interpolation or similar continuous extension, allowing gradients to flow into the table entries $\mathbf{T}$.

**LTCNN Complexity and Gradient Flow:** By increasing the arity of the nodes (typically $n \in \{3, \ldots, 6\}$), the LTCNN reduces the graph depth significantly compared to the LGCNN. For example, with $n = 4$, the path length is halved compared to the binary case ($\log_4 N = 0.5 \log_2 N$). This topological difference theoretically improves gradient signal preservation and reduces the number of sequential non-linearities required to approximate complex Boolean functions over the receptive field.

# F   STATEMENT ON THE USE OF LARGE LANGUAGE MODELS

When producing the text in this publication, Large Language Models (LLMs) were used as a writing tool to clean and refine the authors' original text. Specifically, while the text was written primarily by the lead author, LLMs were used to condense the verbiage of author-written figure captions, identify grammatical errors, and check the clarity of mathematical notation.

