# OpenReview forum: "Exploring weightless neural networks: From logic gates to convolutional lookup tables"
_ICLR.cc/2026/Conference — Submitted to ICLR 2026_

### Official Review · Reviewer_KAEm · 2025-10-20

**Soundness:** 3
**Presentation:** 3
**Contribution:** 2
**Rating:** 6
**Confidence:** 4

**Summary:**

This paper conducts an extensive empirical study on Weightless Neural Networks (WNNs), particularly the Logic Gate Networks (LGNs) and Look-Up-Table Networks (LTNs), exploring their scalability, robustness, and training efficiency relative to standard MLPs and CNNs. The authors introduce a convolutional variant, the LTCNN, designed to mimic CNN kernels via sliding-window logic, and evaluate it against existing LGCNNs.

Three systematic studies are presented:

1. Model Scaling: Comparing training time, accuracy, and noise robustness across model sizes.
2. Bit-Depth Variation: Assessing how quantization granularity (1-, 2-, 4-bit) affects performance.
3. Learnable Mappings: Investigating the impact of learnable interconnects between logic layers.

Results across MNIST, Fashion-MNIST, and CIFAR-10 show that WNNs achieve comparable accuracy to traditional DNNs on simple datasets but require larger parameter counts and training time. LGNs display superior robustness to salt-and-pepper noise, while LTNs generally train faster. However, scaling beyond modest architectures remains challenging due to combinatorial training complexity and limited receptive fields.

**Strengths:**

*Unprecedented experimental scale:* Over 3000 model variations evaluated across architectures, datasets, and encoding schemes — the largest comparative WNN study to date.

*Methodological clarity:* Parameter search, optimization settings, and training details are exhaustively documented, ensuring reproducibility.

*Architectural innovation:* Introduction of LTCNNs extends WNN applicability to spatially structured data.

*Balanced analysis:* Includes multiple performance metrics — accuracy, training time, and robustness — not just raw accuracy.

*Hardware relevance:* Considers inference efficiency for FPGA deployment, highlighting edge-device applicability.

**Weaknesses:**

*Limited conceptual novelty:* Despite broad experimentation, the contribution is primarily empirical — no new training paradigm or theoretical framework is proposed.

*Underdeveloped scalability discussion:* The paper identifies training inefficiency but doesn’t analyze why gradient-based optimization underperforms with discrete structures.

*Missing SOTA comparisons:* Lacks benchmarks against Binary Neural Networks (BNNs) or quantized models (e.g., XNOR-Net, DoReFa-Net), which target similar hardware-efficient goals.

*Overemphasis on small datasets:* Evaluation restricted to MNIST-family and CIFAR-10 — too elementary for claims of “real-world scalability.”

*Ambiguous bit-depth insights:* The bit-depth study’s findings (“depends more on dataset than model”) feel descriptive rather than explanatory.

*Unclear path forward:* Future work is listed but not tied to the limitations uncovered, weakening the narrative closure.

**Questions:**

*Detailed Analyses:*

This paper stands at the crossroads of symbolic determinism and differentiable learning. It is not just a technical benchmark but a philosophical probe into how much “logic” can live inside a modern neural framework.

The study’s brilliance lies in revealing that weightlessness is not simplification — it’s structure exposed. The very mechanisms that make LGNs interpretable — fixed binary operators and explicit mappings — also constrain their ability to scale. This is the paradox of discrete differentiability: transparency breeds rigidity.

Yet, the work’s contribution is not diminished by its empirical focus. It charts the limits of current WNNs while providing an honest, data-driven narrative of their trade-offs. It implicitly calls for hybridization — integrating logic-based regularization or attention-like symbolic layers into conventional deep nets.

In short, the paper answers a deeper question: where do Boolean ideals meet the entropy of gradient descent? And in that meeting, it maps the next horizon of neurosymbolic research.
While not theoretically groundbreaking, this paper’s scale, rigor, and insight make it a valuable empirical cornerstone for the neuro-symbolic community. Its clarity and reproducibility elevate it beyond a routine benchmarking effort. However, it would benefit from stronger engagement with recent SOTA baselines and a more principled discussion of why weightless architectures hit their current limits.

I expect the authors to defend or rebut the points in the weakness section during the rebuttal phase.

---

> ### Author Response · Authors · 2025-11-26
>
> We thank the reviewer for the thoughtful comments. We are glad that the reviewer appreciates the conceptual framing and empirical rigor of our study. We also appreciate the recognition of the interpretability-versus-scalability trade-off in logic-gate networks (LGNs) and the broader perspective on neurosymbolic research. Please note we respond directly to the mentioned weaknesses as requested in the question section.
>
> **Weakness A:** Limited conceptual novelty: Despite broad experimentation, the contribution is primarily empirical — no new training paradigm or theoretical framework is proposed.
>
> - **Response A:** *While the work does not introduce a new training framework, its contribution lies in conducting the first systematic, multi-dimensional empirical assessment of differentiable WNNs. By isolating where current architectures fail, such as instability during optimization, reduced robustness at scale, and sensitivity to bit depth, the study provides an empirical foundation that is necessary before new theoretical advances or paradigms can be meaningfully developed and evaluated.*
>
> **Weakness B:** Underdeveloped scalability discussion: The paper identifies training inefficiency but doesn’t analyze why gradient-based optimization underperforms with discrete structures.
>
> - **Response B:** *We agree with the reviewer’s observation. To address this, we performed an additional analysis by visualizing the loss landscapes of each model, highlighting changes in smoothness, variability, and scale. This comparison shows that gradient magnitudes are generally lower for logic-gate networks, and the loss surfaces for both logic-gate and lookup table networks are significantly less smooth than those of MLPs. These findings help explain why gradient-based optimization struggles with discrete structures, and will be used to further develop statements in the future works section.*
>
> **Weakness C:** Missing SOTA comparisons: Lacks benchmarks against Binary Neural Networks (BNNs) or quantized models (e.g., XNOR-Net, DoReFa-Net), which target similar hardware-efficient goals.
>
> - **Response C:** *We appreciate the suggestion and agree that these models are important hardware-efficient architectures. We did not include them for several reasons. Expanding the study to BNNs and quantized models would broaden the scope, as our analysis already covers over 4,000 trained WNN architectures (per the updates in the coming resubmission). Our focus is specifically on differentiable WNNs, which are novel and require targeted evaluation of their optimization and scalability. Prior studies have compared LGN and LTN models against BNNs, so reproducing those results would be largely duplicative. To acknowledge their relevance, we will include discussion situating BNNs and quantized networks within the broader landscape of efficient architectures and clarifying how their training paradigms differ from differentiable WNNs.*
>
> **Weakness D:** Overemphasis on small datasets: Evaluation restricted to MNIST-family and CIFAR-10 — too elementary for claims of “real-world scalability.”
>
> - **Response D:** *We clarify that smaller datasets were chosen because current WNN training times make large-scale experimentation infeasible. The study should therefore be viewed as an analysis of scalability indicators, such as robustness trends, architecture depth, and bit-depth behavior, rather than a claim of real-world readiness. This framing makes explicit that the goal is to identify limitations that must be addressed before larger-scale evaluation becomes possible.*
>
> **Weakness E:** Ambiguous bit-depth insights: The bit-depth study’s findings (“depends more on dataset than model”) feel descriptive rather than explanatory.
>
> - **Response E:** *We agree that the original bit-depth results were primarily descriptive, and we now provide a clearer explanation using an additional sensitivity analysis. By examining d(Accuracy)/d(Bit-depth) across architectures and model sizes, we show how the MLP, LTN, and LGN model types leverage increased bit-depth information to varying degrees. This analysis clarifies the underlying differences in how each architecture responds to changes in numerical precision.*
>
> **Weakness F:** Unclear path forward: Future work is listed but not tied to the limitations uncovered, weakening the narrative closure.
>
> - **Response F:** *We revise the future-work section to link each proposed direction directly to observed limitations. Kernel optimization addresses the dominant training-time bottleneck, though more research is needed. Alternative architectures and learning strategies should target gradient instability. The new bit-depth studies may be linked to study the model scale needed to leverage certain bit depths. These connections to the included work seeks to establishes a clearer and more actionable research trajectory grounded in our findings.*

---

### Official Review · Reviewer_eTUt · 2025-10-21

**Soundness:** 3
**Presentation:** 3
**Contribution:** 2
**Rating:** 4
**Confidence:** 4

**Summary:**

This paper presents a comprehensive investigation of Weightless Neural Networks (WNNs), specifically Logic Gate Networks (LGNs) and Look-Up Table Networks (LTNs) and compares them to conventional neural models (MLPs and CNNs). It explores a very wide range of model configurations (>3000), analyzing their impact on training time, accuracy, and noise robustness using 3 image-based datasets (MNIST, Fashion-MNIST, CIFAR-10). Impact of learnable mapping (trainable inter-connects) and bit depth of inputs encoding is also studied. As part of their evaluation, the authors also introduce a novel LTN architecture (LTCNN), by applying to LTN the sliding-window mechanism characteristic of LGCNN, an LGN variant.
Results show that, at the range of model sizes investigated, LTNs and LGNs achieve comparable accuracies and noise robustness to their MLP and CNN counterpart, although requiring longer training times. The optimal bit depth is primarily dataset-dependent. Learnable mapping can be beneficial for accuracy but at the cost of significantly increased model size and training time.

**Strengths:**

- The main strength of this paper is the comprehensiveness of the comparative study of LGNs and LTNs, an exploration covering a very wide range of model configurations and test conditions. The results offer a consolidated reference for WNN performance
- The paper is well structured and results are well organized. It is generally easy to follow, although some concepts, such as learnable mapping and sliding-window modification for LGCNN, are taken for granted and not explained for a general audience
- The authors do not overclaim, they offer a balanced discussion of WNNs underperforming/overperforming compared to the counterpart reference models

**Weaknesses:**

- Limited novelty: the primary novelty lies in 1) the introduction of the LTCNN and 2) an extensive configuration sweep. However, LTCNN is conceptually a direct adaptation of the existing LGCNN. It should be noted that the exact sliding window mechanism the authors introduce in LTCNN is not described in details in this paper, although it is understood to be equivalent to the one used in LGCNN. LTCNN do not appear to offer significant performance gains and are slower to train. On the other hand, the broad hyperparameter exploration is not a source of novelty per-se, and the discussion is primarily observational, with speculative explanations
- In terms of impact, a key bottleneck to WNNs practical applicability is the long training time and this paper confirms this limitation rather than offering a solution or mitigation strategy. Consequently, experiments relies on very small-scale image datasets and small models (up to 1M parameters), severely limiting generalization to real-world or large-scale data
- Other explorations (noise robustness, bit width) show mixed results, in the sense that different trends are observed across models. Although interesting, they suffer from the same lack of generalizability to larger datasets and more challenging tasks
- Learnable mappings improve performance but exacerbates the fundamental limitation of WNN, the long training time, further worsening scalability

**Questions:**

- Can LTCNN be optimized to improve training time, similarly to kernel optimization implemented for LGCNN in the cited Petersen et al. 2024?
- Can the observed trends be generalized to more complex datasets, at least in some scenarios like noise robustness?

---

> ### Author Response · Authors · 2025-11-26
>
> Thank you for your insightful review, and for highlighting both the strengths of our comparative study and the areas where further clarification is needed. We have reviewed your comments and conducted additional experiments to address your concerns which will be reflected in the revised manuscript. Please see responses below.
>
> **Question 1:** Can LTCNN be optimized to improve training time, similarly to kernel optimization implemented for LGCNN in the cited Petersen et al. 2024?
>
> - **Response 1:** *Optimizing the LTCNN with a specialized kernel is feasible and would likely improve training time. However, direct comparisons with Petersen et al.’s kernel are currently not possible, as their implementation is unavailable. The LTCNN is primarily introduced to benchmark how differentiable WNNs scale to larger architectures, highlighting the need for improved training routines. Recognizing the benefits, we will discuss the value of optimized kernels in our future work section.*
>
> **Question 2:** Can the observed trends be generalized to more complex datasets, at least in some scenarios like noise robustness?
>
> - **Response 2:** *We are cautious about claiming generalizability to large-scale datasets (e.g., CIFAR-100, ImageNet). For reference, LGCNN achieved ~86% accuracy as of November 2024, and better architectures may be required for larger-scale studies. Instead, we focus on the limitations of model primitives as a precursor to scaling WNNs. Our analysis shows that robustness varies by dataset and model, but LTCNN and LGCNN architectures consistently exhibit larger beta values which indicates brittleness which remains data-agnostic. While future work is needed to confirm these trends on complex datasets, our study provides a benchmark to aid comparison.*
>
> **Weakness A:** Limited novelty: the primary novelty lies in 1) the introduction of the LTCNN and 2) an extensive configuration sweep. However, LTCNN is conceptually a direct adaptation of the existing LGCNN. It should be noted that the exact sliding window mechanism the authors introduce in LTCNN is not described in details in this paper, although it is understood to be equivalent to the one used in LGCNN. LTCNN do not appear to offer significant performance gains and are slower to train. On the other hand, the broad hyperparameter exploration is not a source of novelty per-se, and the discussion is primarily observational, with speculative explanations
>
> - **Response A:** *As a performance comparison study, our focus is on benchmarking the challenges of scaling differentiable WNNs rather than proposing new architectures or theoretical contributions. While not the focus, the LTCNN improves training time by 4× compared to the LGCNN and achieves higher accuracy. However, to address the weakness, the revised manuscript adds several new analyses, including visualization of the loss landscape for LTN and LGN, a deeper investigation of bit depth, data augmentation studies, and experiments comparing varied training strategies. The LTCNN architecture will be better described.*
>
> **Weakness B:** In terms of impact, a key bottleneck to WNNs practical applicability is the long training time and this paper confirms this limitation rather than offering a solution or mitigation strategy. Consequently, experiments relies on very small-scale image datasets and small models (up to 1M parameters), severely limiting generalization to real-world or large-scale data
>
> - **Response B:** *We agree that long training times are a key limitation of WNNs and a barrier to scalability. Our study emphasizes this limitation as a central message. Rather than claiming immediate real-world applicability, our work categorizes the challenges preventing deployment and establishes a benchmark to guide future research toward more scalable WNNs.*
>
> **Weakness C:** Other explorations (noise robustness, bit width) show mixed results, in the sense that different trends are observed across models. Although interesting, they suffer from the same lack of generalizability to larger datasets and more challenging tasks
>
> - **Response C:** *While trends vary across datasets and model sizes, general patterns emerge: feed-forward models are generally more robust than CNN counterparts, and bit depth studies reveal that appropriate settings depend on dataset characteristics. To enhance this analysis, we will include a sensitivity study of model accuracy with respect to bit depth ("dA/dBit-Depth"), which provides more general insights across model types.*
>
> **Weakness D:** Learnable mappings improve performance but exacerbates the fundamental limitation of WNN, the long training time, further worsening scalability
>
> - **Response D:** *We agree that learnable mappings highlight a trade-off. They improve accuracy but exacerbate the scalability challenge. Our study emphasizes this limitation and provides detailed insights, highlighting the need for more efficient strategies to learn interconnections in WNNs.*

---

### Official Review · Reviewer_czrz · 2025-10-23

**Soundness:** 2
**Presentation:** 1
**Contribution:** 2
**Rating:** 2
**Confidence:** 4

**Summary:**

This paper presents an empirical comparison of Weightless Neural Networks (WNNs)—Logic Gate Networks (LGNs) and Look-Up Table networks (LTNs)—against traditional deep neural networks (MLPs and CNNs). The authors train 1040+ model architectures across MNIST, Fashion-MNIST, and CIFAR-10 to evaluate test accuracy, training time, and robustness to noise.

**Strengths:**

Training 1040 architectures across three datasets with multiple evaluation dimensions (accuracy, training time, robustness) represents a significant experimental effort.

Introduction of convolutional LTN variants fills a gap in the literature and enables fair comparison with LGCNNs.

The paper addresses real engineering questions (training time, robustness, bit depth) relevant to practitioners considering WNN deployment.

Beyond accuracy, the robustness analysis (salt-and-pepper noise, occlusions) and training time measurements provide valuable practical insights.

The paper also tests Fashion-MNIST, which was not done by Petersen et al. (2022;2024), and something that was missing in their evaluation.

**Weaknesses:**

The statement “In real-world deployments, applying augmentation would likely improve performance” should simply be tested.

The paper's core motivation is FPGA deployment and inference speed, yet never measures either.
All experiments run on GPUs (NVIDIA L4)
No inference time measurements reported
No hardware resource utilization (LUTs, power consumption)
No comparison to actual FPGA implementations
Some or all of these are critical to draw the real-world conclusions the authors do.

The statement “Note that LGNs and LTNs achieve state-of-the-art performance for MNIST and Fashion-MNIST (i.e. hand written characters and clothing items) while performing worse on CIFAR-10 (i.e. containing structurally complex images of birds, cars, and other classes), allowing these datasets to stress each model’s performance and reveal challenges with training complex model architectures” requires citations.

No error bars on accuracy measurements despite stochastic training

2-fold validation is unusual—why not standard 80/10/10 or 5-fold cross-validation?
Averaging over "top 5 models" biases results toward best-case scenarios

Some missing related work. A few of these are merely concurrent work, but it makes sense to cite given the overlap.
https://arxiv.org/abs/2508.17512
https://arxiv.org/abs/2506.07500 (you already mean to cite this. It is the Yousefi & Wattenhofer 2025 citation)
https://ieeexplore.ieee.org/document/10301592
https://arxiv.org/abs/2510.03250
https://arxiv.org/abs/2506.04912
https://arxiv.org/abs/2509.25933
https://arxiv.org/abs/2504.00592

You cite “Shakir Yousefi and R Wattenhofer. Deep differentiable logic gate networks: Neuron collapse through a neural architecture search perspective. 2025.” However, this is a project description. Yousefi published their work in the Mind the Gap paper (https://arxiv.org/abs/2506.07500).

Captions for the tables should be above the tables as per the formatting instructions.

The figures are generally low resolution, and the font is small. Please address this.

The story of the paper is interesting; however, the writing is rather clunky, and the presentation could be improved. This is in particular the case for sections 3.2 and 4.

The color-coded bars in Tables 2 and 3 are hard to interpret.

Typos:
Line 50 should have “ML” rather than “Ml.”


While my review is rather negative, the authors can and should address several of these things for the iclr submission, as the paper and reviews will be public. The issues with the citations, missing citations, figures, captions, etc., can be resolved within a day :)

**Questions:**

What is the training and validation split? Line 187 makes it sound like 50/50, but this seems quite aggressive.

Why did you leave out all data augmentations? I understand omitting some to test generalization to unseen perturbations; however, if you want to determine their real-world applicability, then this should be included.

Why did you use the quantization method over a temperature encoding as Petersen et al. use?

What is the full distribution of accuracies (not just top-5)?

Can you create a figure for section 3.2, as it is currently not easy to understand?

How many layers do the models have? (e.g. in Table 1).

Do you know why LTCNN’s time per epoch drops a lot for the largest models in Table 1. Both the mean and the std are very low.

Why are your CNN on CIFAR results so poor? It should not be hard to get an accuracy around 80% (https://www.kaggle.com/code/faressayah/cifar-10-images-classification-using-cnns-88)

You report test accuracies for your DWN that are much lower than in the original DWN paper. Your Table 3 Fashion MNIST test accuracies are around 55% while the DWN paper reports 89% (see Table 1 https://arxiv.org/pdf/2410.11112). Why is this?

---

> ### Author Response · Authors · 2025-11-26
> **Rebuttal 1 [Part 1]**
>
> Thank you for providing a clearly thoughtful review and for recognizing the experimental effort and the pursuit of practical outcomes. We value the opportunity to address your feedback, and have pursued additional work to address your concerns which we are excited to include in a revised submission. Please see responses below:
>
> **Question 1:** What is the training and validation split? Line 187 makes it sound like 50/50, but this seems quite aggressive.
>
> - **Response 1:** *A split of 50/50 was used for the original submission. This is aggressive, and was performed so the experiments could complete prior to submission. At this time, all experiments have been re-run using 5-Fold validation which will be presented in the revised submission. The experimental findings remain largely unchanged.*
>
> **Question 2:** Why did you leave out all data augmentations? I understand omitting some to test generalization to unseen perturbations; however, if you want to determine their real-world applicability, then this should be included.
>
> - **Response 2:** *As observed, excluding data augmentation aids tests to unseen perturbations. Here, three properties hold value: (1) a model's **innate robustness,** having never seen noise; (2) a model's **trained robustness,** having seen noise during training; (3) the **delta-robustness,** or the difference between 1 and 2. Both innate and trained robustness capture a model's performance when observing missing information. However, innate robustness tests noise outside the training distribution, while trained robustness shifts the training distribution to overlap with noisy samples. The delta-robustness shows how model's can learn to make predictions despite missing information. Both expected and unexpected types of noise may be observed in the real-world and hold value (though many anomalies types exist). Recognizing the weakness in only capturing the innate robustness, we will include an additional experiment with data augmentations to study the trained robustness and delta-robustness.*
>
> **Question 3:** Why did you use the quantization method over a temperature encoding as Petersen et al. use?
>
> - **Response 3:** *We use quantization rather than temperature encoding to reduce the number of inputs used for training (e.g. quantization and temperature encodings requires 2-bits and 3-bits to represent 4 values, respectively). This approach recognizes that reducing bit depth alleviates bandwidth constraints for real-world applications, and allows our architecture search method to discover more model architectures which satisfy our constraints for small model sizes (e.g. the first layer of logic-gate networks should use (# gates) >= (0.5)x(# inputs), which is infeasible when using larger bit depths for images).*
>
> **Question 4:** What is the full distribution of accuracies (not just top-5)?
>
> - **Response 4:** *The full set was not included to avoid cluttering the visuals. The updated submission will include statistics covering the full distribution of model accuracies.*
>
> **Question 5:** Can you create a figure for section 3.2, as it is currently not easy to understand?
>
> - **Response 5:** *We are glad to include a figure. Figure 1 will be updated to save space.*
>
> **Question 6:** How many layers do the models have? (e.g. in Table 1).
>
> - **Response 6:** *Table 4 includes the number of model layers per model architecture, which varies from 1-8 for CNN-based architectures, and from 1-16 for feed-forward models. This table is currently in the appendix to save space.*
>
> **Question 7:** Do you know why LTCNN’s time per epoch drops a lot for the largest models in Table 1. Both the mean and the std are very low.
>
> - **Response 7:** *The low time per epoch reported for the LTCNN was an error in analysis. The full set of model training times is updated per the repeated experiment discussed in from Response 1, where the new table represents the time-per-batch so results better represent datasets with variable number of samples.*
>
> **Question 8:** Why are your CNN on CIFAR results so poor? It should not be hard to get an accuracy around 80% (https://www.kaggle.com/code/faressayah/cifar-10-images-classification-using-cnns-88)
>
> - **Response 8:** *We identified several factors contributing to low CNN accuracy on CIFAR in the original experiments: (1) the learning rate was set too high (0.2), resulting in negligible or unstable learning, (2) our CNN architectures were sparse to maintain comparability with LTCNN and LGCNN models, and (3) we excluded batch normalization and dropout to ensure architectural consistency and to reduce confounding factors. After adjusting the learning rate to 0.001, CNN performance improved to 62.4%. Differences from the ~87% reported in the Kaggle reference are expected due to sparsity, absence of normalization, and random sampling of architectures rather than designing a hand-crafted architecture.*

---

> > ### Author Response · Authors · 2025-11-26
> > **Rebuttal 1 [Part 2]**
> >
> > **Question 9:** You report test accuracies for your DWN that are much lower than in the original DWN paper. Your Table 3 Fashion MNIST test accuracies are around 55% while the DWN paper reports 89%... Why is this?
> >
> > - **Response 9:** *With thanks for the comparison and in review of the claim, we find comparable results between the original paper and our submission. The original authors achieve an 89\% accuracy, for a model with 7.8KiB parameters. Here, we may convert the number of parameters in bytes to a comparable size range (7.8 KiB = 7.8 × 1024 bytes of 1-bit parameters = ~63,898 parameters = ~$10^{4.8}$). Looking at Table 3 from our initial submission, the LTN achieves am accuracy of 82-83\% for a slightly smaller model trained with ~$10^{4.5}$ parameters. Here, the discrepancy of 6\% may come from (1) the smaller model size, and (2) the use of learnable mappings in the original work, as our results in Table 3 did not use learnable mappings.*
> >
> > **Weakness A:** The statement “In real-world deployments, applying augmentation would likely improve performance” should simply be tested.
> >
> > - **Response A:** *Please see Response 2 regarding an additional augmentation study.*
> >
> > **Weakness B:** The paper's core motivation is FPGA deployment and inference speed, yet never measures either. All experiments run on GPUs (NVIDIA L4) No inference time measurements reported No hardware resource utilization (LUTs, power consumption) No comparison to actual FPGA implementations Some or all of these are critical to draw the real-world conclusions the authors do.
> >
> > - **Response B:** *We acknowledge this concern. While our study does not directly characterize FPGA performance, we intentionally focus on model training, which represents a primary bottleneck for deploying WNNs. To address this limitation, we will revise the manuscript to clarify the scope and highlight device-specific considerations, including the challenges of deployment on edge devices such as FPGA timing constraints, resource availability, and model size limitations.*
> >
> > - *The motivation for FPGA and edge deployment remains valid, as prior work demonstrates that WNNs can improve inference speed, reduce resource usage, and lower power consumption (e.g. https://arxiv.org/html/2411.04732v1). However, practical deployment involves multiple additional challenges, including: (1) achieving sufficient model accuracy through training, (2) fitting models within device-specific resource constraints, and (3) meeting timing requirements for low-latency applications. A comprehensive study of all these factors is beyond the scope of this work, which focuses on benchmarking model training and scalability.*
> >
> > - *Focusing on training has several advantages: (1) it addresses a primary bottleneck in WNN deployment, (2) it remains hardware-agnostic and broadly applicable across devices, and (3) it allows comparison to current research trends in differentiable logic-gate networks (LGNs) and look-up-table networks, which have largely been limited to small models and simple datasets (e.g. https://arxiv.org/abs/2210.08277). By prioritizing training, our work highlights key scaling challenges and provides a foundation for future studies targeting efficient deployment on FPGAs and other edge devices.*
> >
> > **Weakness C:** 2-fold validation is unusual—why not standard 80/10/10 or 5-fold cross-validation?
> >
> > - **Response C:** *The experiment was rerun with 5-Fold cross validation. Please see Response 1.*
> >
> > **Weakness D-F:** (1) The statement... requires citations. (1) Some missing related work. A few of these are merely concurrent work, but it makes sense to cite given the overlap... (2) You cite... However, this is a project description. Yousefi published their work in the Mind the Gap paper...
> >
> > - **Response D-F:** *Several references were published near the original submission deadline. With the reviewer’s guidance, we have now added the relevant citations in the revised manuscript.*
> >
> > **Weakness G-M:** (1) Averaging over "top 5 models" biases results toward best-case scenarios. (2) No error bars on accuracy measurements despite stochastic training. (3) Captions for the tables should be above the tables... (4) The figures are generally low resolution, and the font is small... (5) The story of the paper is interesting; however, the writing is rather clunky... This is in particular the case for sections 3.2 and 4. (6) The color-coded bars in Tables 2 and 3 are hard to interpret. (7) Typos: Line 50 should have “ML” rather than “Ml.”
> >
> > - **Response G-L:** *We have addressed all presentation and clarity issues. Figures and tables will be updated, captions moved above tables per formatting instructions, resolution and font sizes improved, and color-coding clarified. Top-5 model reporting will be replaced with full distribution statistics, and typographical errors will be corrected. Sections 3.2 and 4 will be restructured for improved readability, ensuring the narrative is clearer and easier to follow.*

---

### Author Response · Authors · 2025-12-03
**Author Final Remarks**

Dear Area Chairs and Program Chairs,

Thank you for your continued effort in managing the review process following the OpenReview API security incident. Because our revised experiments were still in progress prior to the incident, we were unable to submit our rebuttal revision or receive reviewer feedback on the updated results. To support proper evaluation, we summarize the full set of changes made in response to the reviewers’ requests.

**Addressing Anomalous Experimental Design:**
- All experiments were fully repeated using 5-fold cross-validation, replacing the previous 50/50 split and resolving concerns from reviewers czrz. All results now report full distribution statistics rather than top-k subsets, and error bars are added across Figures 1 through 8 and Tables 1 through 6. Previously anomalous values were corrected, such as the LTCNN timing error (czrz) and the low CIFAR-CNN baseline.


**Addressing Novelty, Experimental Depth, and Requested Analyses:**
- To address critiques regarding novelty and experiment completeness (KAEm, eTUt), we significantly expanded Sections 2 through 4. Specifically, we added: (1) a loss-landscape visualization illustrating optimization challenges in discrete kernel architectures (KAEm); (2) a bit-depth sensitivity analysis, computing d(Accuracy)/d(bit-depth) to quantify how each architecture leverages precision (KAEm); (3) a data-augmentation experiment (czrz); and (4) a study comparing encoding schemes to situate quantization and thermometer-style encodings within the broader precision-efficiency trade-off. We also strengthen our novelty statements in Section 2, clarified the contributions of DWNN convolutions, and expanded the contextual comparison to BNNs and quantized CNNs. We emphasize that our work is positioned as an analysis of scalability indicators, not a claim of immediate real-world deployment readiness.

**Addressing Missing LTCNN Architectural Description:**
- Per request (eTUt), we added a complete appendix detailing the LTCNN architecture, including an overview of LUT-based convolution and distinctions from the LGCNN.


**Addressing Writing Quality, Figure Interpretability, and Presentation Issues:**
- We rewrote Sections 3.2 and 4 for clarity, readability, and improved connection between motivation, results, and conclusions (czrz, KAEm). All figures were updated with improved resolution and font sizes; captions for tables were moved above the tables; and several previously confusing figures were replaced with tables to better communicate distribution-level results. Citations were corrected or added as requested, and the methodology section was revised to integrate the newly added experiments into a coherent narrative.


**Addressing Scope, Hardware Clarifications, and Future Work:**
- Responding to comments on scope (KAEm, eTUt, czrz), we clarified that the central bottleneck for DWNN scaling is training complexity, which precedes hardware or FPGA deployment considerations. We further revised the framing of the paper to emphasize analytical insights and benchmarking rather than proving practical readiness. The future works were rewritten so that each proposed direction is connected to a paragraph in the results section. Finally, to support reproducibility and add novelty, we plan to release the full experimental software as the first open-source DWNN convolutional library upon acceptance; the repository remains private to preserve anonymity.

In summary, we believe this work adds significant value to ICLR by offering an essential empirical foundation for the neurosymbolic and efficient ML communities. By conducting the first systematic, large-scale assessment of differentiable weightless neural networks (i.e. benchmarking over $\text{4,000}$ model configurations) we have quantified key performance trade-offs and provided clear directions for continued research. Our findings draw necessary attention to the challenges that must be overcome to realize power-efficient ML solutions at scale, which holds the potential to significantly reduce the power demands, circuit area, and inference times of modern ML architectures for on-edge devices and efficient ASIC designs.

We hope these revisions clarify our contributions and fully address reviewer concerns. Thank you for your time and consideration.

---

### Meta-Review · Area_Chair_eRZs · 2026-01-06

**Summary:**

This paper presents a large scale empirical study of differentiable weightless neural networks, including Logic Gate Networks and Look Up Table Networks, with extensive comparisons to MLPs and CNNs across accuracy, training time, and robustness. Reviewers consistently recognize the unprecedented experimental breadth, spanning thousands of configurations, and the introduction of a convolutional LUT variant as valuable contributions. At the same time, the work is primarily empirical in nature, with limited conceptual novelty, and it highlights rather than resolves key scalability challenges of weightless models, particularly long training times and increasing brittleness at larger model sizes.

**Reviewer Concerns:**

Most substantive reviewer concerns were addressed in the revision. The authors reran all experiments using five fold cross validation, added full distribution statistics and error bars, corrected experimental anomalies, expanded analyses such as loss landscape visualization, bit depth sensitivity, data augmentation, and encoding comparisons, clarified the LTCNN architecture, and substantially improved writing and presentation quality. Remaining concerns are largely structural rather than technical, including limited novelty relative to prior logic gate convolutions, reliance on small datasets and modest model scales, and the absence of direct hardware or FPGA level measurements. These limitations are now clearly acknowledged and appropriately framed as open challenges motivating future work.

**Reviewer Scores:**

Reviewer opinions remain mixed but cluster around the borderline. One reviewer rated the paper marginally above the acceptance threshold, citing the scale, rigor, and insight of the empirical study, while others remained at marginal reject despite acknowledging substantial improvements.

---

### Decision · Program_Chairs · 2026-01-26

Reject